# Single-cell gene fusion detection by scFusion

Zijie Jin[1], Wenjian Huang [2], Ning Shen [3,4], Juan Li [5], Xiaochen Wang[1], Jiqiao Dong[6], Peter J. Park [4] &
Ruibin Xi [1,7 ✉]

Gene fusions can play important roles in tumor initiation and progression. While fusion detection so far has been from bulk samples, full-length single-cell RNA sequencing (scRNA-seq) offers the possibility of detecting gene fusions at the single-cell level. However, scRNA-seq data have a high noise level and contain various technical artifacts that can lead to spurious fusion discoveries. Here, we present a computational tool, scFusion, for gene fusion detection based on scRNA-seq. We evaluate the performance of scFusion using simulated and five real scRNA-seq datasets and find that scFusion can efficiently and sensitively detect fusions with a low false discovery rate. In a T cell dataset, scFusion detects the invariant TCR gene recombinations in mucosal-associated invariant T cells that many methods developed for bulk data fail to detect; in a multiple myeloma dataset, scFusion detects the known recurrent fusion *IgH-WHSC1*, which is associated with overexpression of the *WHSC1* onco-gene. Our results demonstrate that scFusion can be used to investigate cellular heterogeneity of gene fusions and their transcriptional impact at the single-cell level.

[1] School of Mathematical Sciences, Peking University, Beijing 100871, China. [2] Academy for Advanced Interdisciplinary Studies, Peking University, Beijing 100871, China. [3] Liangzhu Laboratory, Zhejiang University Medical Center, Hangzhou 311121, China. [4] Department of Biomedical Informatics, Harvard Medical School, Boston 02115 MA, USA. [5] Department of Biomedical Engineering, College of Engineering, Peking University, Beijing 100871, China. [6] GeneX Health Co. Ltd, Beijing 100195, China. [7] Center for Statistical Science, Peking University, Beijing 100871, China. ✉email: ruibinxi@math.pku.edu.cn

Gene fusions are formed by juxtaposition of parts of two genes, resulting from structural rearrangements such as deletions and translocations[1]. In cancer cells, many gene fusions are driver mutations that play important roles in carcinogenesis. Well-known examples include *BCR-ABL1* in chronic myeloid leukemia[2], *TMPRSS2–ERG* in prostate cancer[3], and the *ALK* fusions in lung cancer[4]. Some gene fusions are strongly correlated with tumor subtypes and are often used as diagnostic markers for malignancy. Gene fusions are also important targets for cancer drugs. A number of such drugs have been approved by the US Food and Drug Administration such as imatinib targeting *BCR-ABL1*[5], crizotinib targeting *ALK* fusions[6], and the recently approved TRK inhibitor larotrectinib targeting *NTRK* fusions[7].

RNA sequencing (RNA-seq) provides an accurate and unbiased platform for gene fusion detection. Gene fusions generate chimeric reads that cover junctions of independent partner genes as well as discordant read pairs whose mates map to the two sides of the junction. After mapping, gene fusions can be detected based on a combination of these split and discordant reads to the partner genes. Gene fusions can also be detected from DNA sequencing data, especially for fusions that are difficult to detect from RNA-seq due to their low expression.

Single-cell RNA sequencing (scRNA-seq)[8–10] technologies have transformed our understanding of transcriptional heterogeneity in tissues. Numerous algorithms have been developed to quantify expression levels, identify clusters of cells belonging to the same cell types, and identify developmental trajectories of single cells[11]. With the greater coverage along the length of each transcript by recent protocols, another potential application of scRNA-seq is the detection of gene fusions at the single-cell level. Such analysis could help to identify cell subtypes or subclones in which a fusion plays a role and quantify its impact on transcriptomic output. Although many computational fusion detection tools have been developed for bulk RNA-seq data[12–20], fusion detection using scRNA-seq data is still challenging due to the following: (1) the heavy amplification step may generate artificial chimeric reads, leading to false-positive fusion candidates; (2) the power for detecting fusions shared among multiple single cells can be improved if all single cells are jointly analyzed; (3) current single-cell data often contain thousands or more single cells, making the total size of a collection of cells large, even though the data size for each cell is substantially smaller than that of a bulk RNA-seq sample.

In this paper, we describe a gene fusion detection algorithm named scFusion for scRNA-seq data. scFusion employs a statistical model and a deep-learning model to control for false positives. To assess its performance, we first simulated single-cell data based on real scRNA-seq data and found that scFusion achieved high sensitivity while maintaining a low false discovery rate (FDR). Next, we introduced spike-in fusions experimentally to single cells and validated that those fusions could be detected successfully. Finally, we applied scFusion to four publicly available scRNA-seq data and showed that scFusion identified cell subtypes closely associated with gene fusions.

## Results

**scFusion detects gene fusions using statistical and machine learning models.** scFusion takes as input reads mapped by STAR[21] (Fig. 1). Following a standard procedure for fusion detection in bulk RNA-seq, unique split-mapped reads and discordant reads mapped to different genes are identified and clustered to obtain a candidate gene fusion list. After filtering fusion candidates in pseudogenes, long noncoding RNAs (lncRNAs), genes without approved symbols[22] (such as *RP11-475J5.6*), and in the intronic regions, we observed well over 10,000 fusion

candidates in most datasets tested (see the following sections). When considering only candidates found in at least two cells, there were still thousands of fusion candidates, nearly all of which are likely to be false positives. For example, the T cells of a hepatocarcinoma scRNA-seq dataset[23] had ~1400 fusion candidates shared in at least two cells (excluding candidates involving T cell receptor-related genes). Since these T cells are normal cells, true fusions should be very rare in this dataset. To control for false discoveries, scFusion applies a statistical model and a deep-learning model to filter the potential false positives ("Methods"). Subsequently, scFusion applies two more filters to filter potential false positives that are likely generated by incorrect alignments of short reads (e.g., reads from genes with homologous sequences). If the number of supporting discordant reads for a fusion is more than 10 times that of its supporting split-mapped reads, scFusion filters the fusion candidate. If a gene is in more than five fusion candidates, we filter all fusion candidates involving this gene. The lncRNA and no-approved-symbols filters are optional and users can choose to disable these filters.

There are two assumptions in the statistical model: (1) the candidate fusion list only contains a very small fraction of true fusions, and (2) the true fusion transcripts generally should have more supporting cells/reads than those arising from various experimental and computational artifacts. The distribution of supporting reads that arise due to this background technical noise introduced during sequencing and mapping can be estimated using all candidate fusions. True fusions can be identified if their supporting reads are much larger than those that could be observed from the background distribution. We model the background distribution using the zero-inflated negative binomial (ZINB) distribution. Compared with the Poisson or negative binomial distributions that are often used in scRNA-seq analysis[24–27], the ZINB distribution can model count data with overdispersion and the excessive number of zeros. The data matrix of supporting chimeric reads has a large number of zeros (>95%) and thus the ZINB model is more suitable. In addition, we observe that, for each fusion candidate, the number of supporting chimeric reads in a cell depends on the expression level of the partner genes and the local GC content (Fig. 2a, b). We describe this dependence using a regression model. Since gene expression is linearly correlated with the supporting read count for the most part (Fig. 2a), the regression functions for the gene expression are chosen to be linear. The GC-content dependency is nonlinear (Fig. 2b), and we use splines to represent this nonlinear relationship. After estimating the parameters of the regression model, we perform statistical tests to determine if the fusion candidates come from the background noise ("Methods").

The statistical model alone is not sufficient to filter out all false fusion candidates that may be induced during the experimental procedure (e.g., cell lysing, library construction, and sequencing). We observed that the sequences near the junctions of the chimeric reads are enriched with sequences like AAAA and AGGT (Supplementary Fig. 1a), and hypothesized that the technical artifacts might be learned by a machine learning model. We therefore used a bi-directional Long Short Term Memory network (bi-LSTM)[28,29] to learn and filter the artifacts ("Methods" and Supplementary Fig. 1b). We set the negative training data as the subsequences from chimeric reads, representing the technical artifacts. Since the number of true fusions is too small for the positive set of the bi-LSTM training, we use as the positive set the sequences generated by concatenating random pairs of short reads ("Methods"). The bi-LSTM assigns each candidate an artifact score from 0 to 1 and candidates with artifact scores greater than 0.75 are filtered.

We first evaluated the performance of the bi-LSTM model using six publicly available scRNA-seq datasets[23,30–34] from

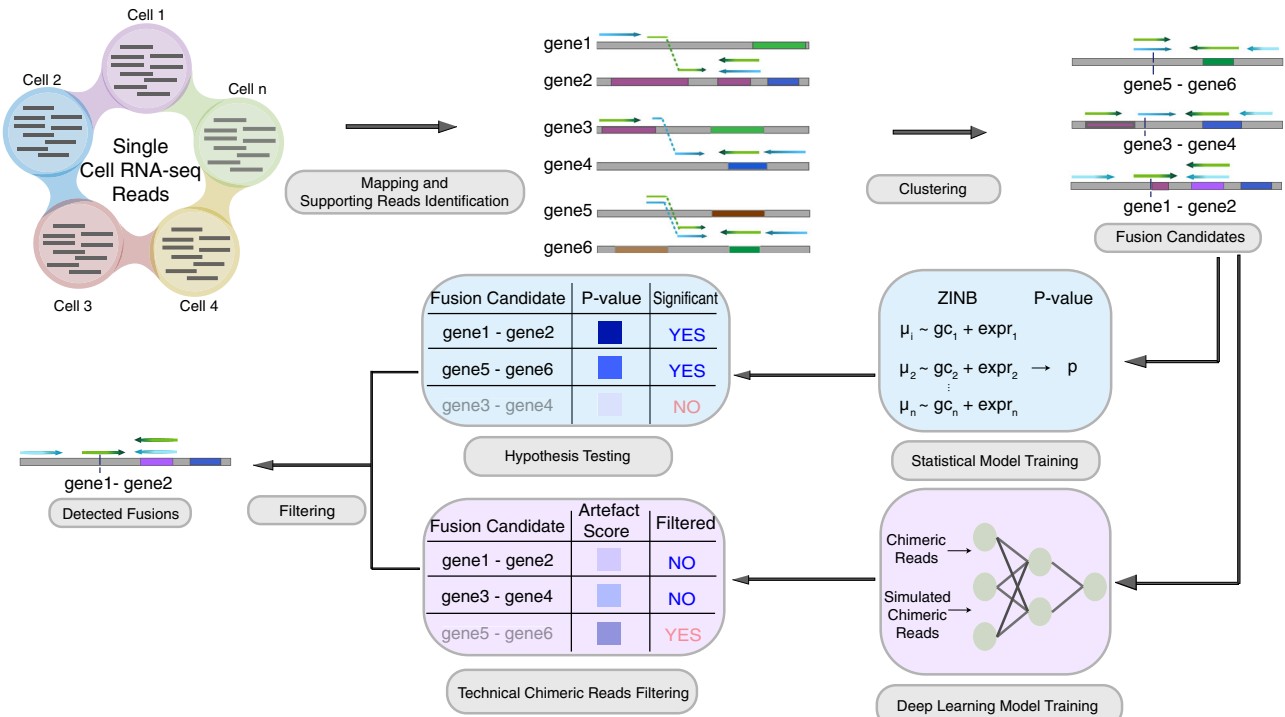

**Fig. 1 Overview of scFusion for single-cell gene fusion detection.** The single-cell RNA-seq reads are mapped and supporting reads are identified and clustered to obtain the fusion candidates list. Given the candidate information, a ZINB-based statistical model and a deep-learning model are trained to filter the potential false positives.

different cancer types. The positive data were the chimeric reads from these data and the negative data were generated as described above. The median of the area under the curve (AUC) was 0.884 and the median area under the precision-recall (PR) curve (AUPR) was 0.913 (Fig. 2c, d). Since the bi-LSTM model was not trained using true fusions, true fusions could be filtered by this model. To evaluate this possibility, we applied the bi-LSTM models trained using six public scRNA-seq data to 3500 gene fusions reported in the Pan-Cancer Analysis of Whole Genomes (PCAWG) studies[35,36]. The PCAWG fusions can be viewed as gold standard true fusion candidates. The PCAWG fusions and the chimeric reads had artifact scores centered around 0.1 (Fig. 2e) and 0.95 (Fig. 2f), respectively. More than 90% of the PCAWG fusions had artifact scores smaller than 0.5, and only ~5% had scores larger than 0.75. This result indicated that the bi-LSTM model could effectively filter chimeric artifacts at the expense of filtering a very small portion of true fusions. Further, the bi-LSTM did learn features of chimeric artifacts. Chimeric reads with high artifact scores can be partially explained by features such as their junction sequences (Supplementary Fig. 2a–e).

**scFusion detects fusions with high sensitivity and precision in simulation.** We compared the performance of scFusion with that of directly applying several popular bulk-sample methods (Arriba[19], STAR-Fusion[20], FusionCatcher[13], and EricScript[18]). We used FusionSimulatorToolkit[20] to generate simulation data with 100 fusions from PCAWG at various expression levels (Supplementary Data 1). Since technical chimeric reads cannot be generated by available simulation tools, we added technical chimeric reads using a method mimicking the mis-priming in PCR amplifications[37] ("Methods"). We set the percentage of chimeric reads to be 1%, similar to that in the real scRNA-seq data (Supplementary Fig. 3a). We varied single-cell numbers (500 and 1000 cells) and data sizes of single cells (2, 3, and 4 million reads,

similar to those in real data; see Supplementary Fig. 3b). In total, we had six different simulation setups and generated ten datasets for each setup. Simulated fusions were randomly added to 20% of cells on average. For bulk methods, we only considered fusions that were reported in at least two cells with at least three supporting reads to reduce their false positives.

Compared with bulk methods, scFusion showed similar levels of recalls but higher levels of precisions and F-scores (Fig. 3, Supplementary Fig. 4, and Supplementary Data 2–6). For example, in the simulation with 1,000 single cells and 4 million reads per cell, the precision and F-score of scFusion are 0.921 and 0.925, respectively, whereas the precisions of the bulk methods are only 0.434–0.503 and the F-scores are 0.574–0.667.

To see the effect of the deep-learning model, we also evaluate the performance of scFusion without the deep-learning model. We find that the deep-learning model greatly helps to improve the precision (the precision increases by 0.049–0.168) and only has a minimal influence on the sensitivity (the sensitivity decreases less than 0.04). The fusions missed by scFusion are mostly fusions with low expression levels (Supplementary Fig. 5).

**Computational efficiency and effects of the filters in real scRNA-seq data.** We considered five scRNA-seq datasets. These include a newly sequenced spike-in dataset (729 cells), a T cell dataset[23] from a liver cancer study (2355 cells), a multiple myeloma dataset[33] (597 cells), and two prostate cancer datasets[31,38] (288 cells and 922 cells, respectively). Details about the data are described below. We first compared the computational time of different algorithms in these datasets (Fig. 4a). The computational time of scFusion is only about a half of Arriba and ~30% or less of STAR-Fusion, EricScript, and FusionCatcher, demonstrating the superior computational efficiency of scFusion. Figure 4b shows the effects of different filters in the scRNA-seq data. The statistical model is responsible for filtering most of the potential false positives.

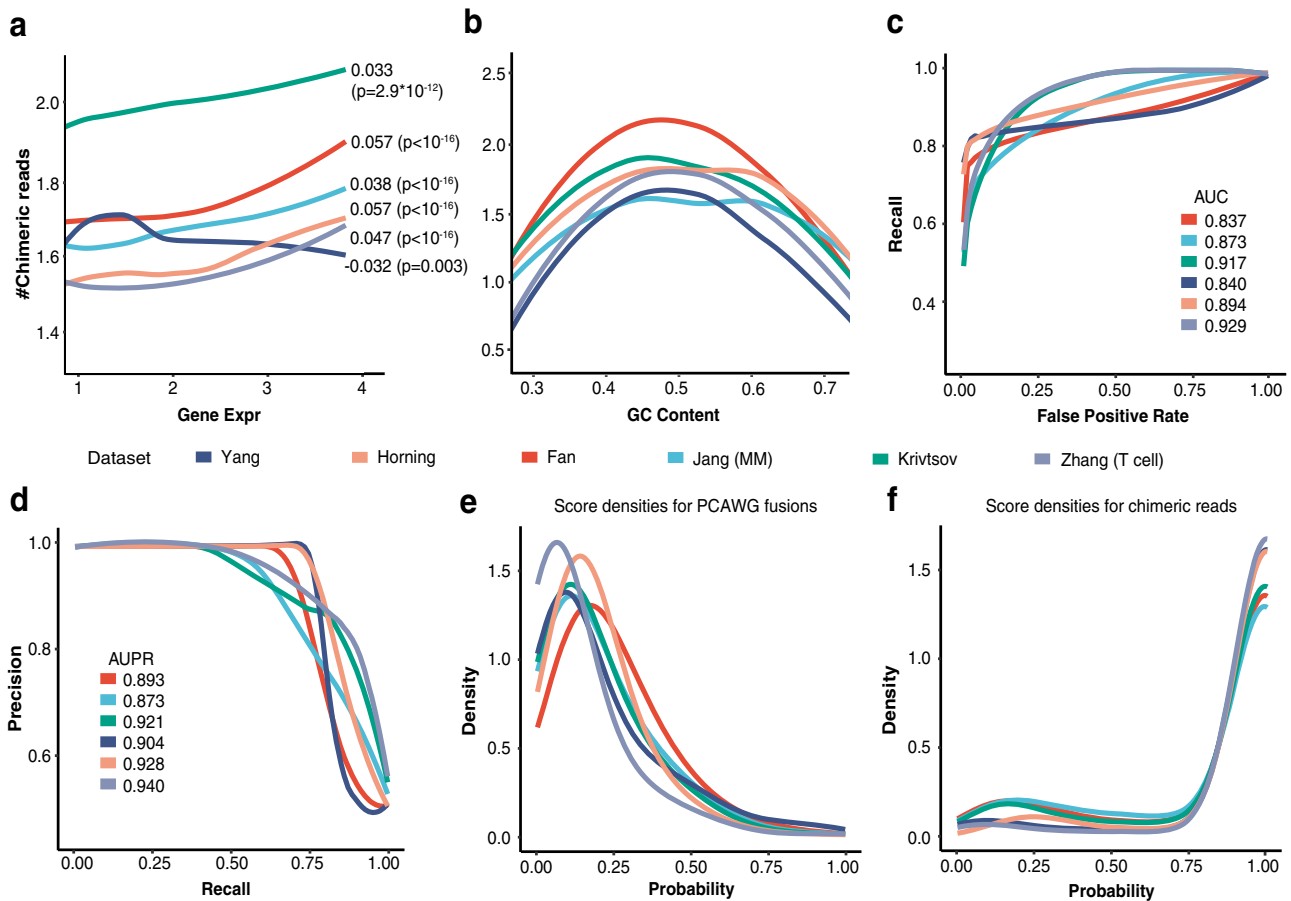

**Fig. 2 Features of technical chimeric reads.** The number of supporting chimeric reads depends on (**a**) the expression of partner genes and (**b**) the local GC content. The Pearson's Correlations between the number of chimeric reads and the gene expression and the p-values are shown in the figure. P-values were calculated by two-sided Student's *t* test. The GC content was calculated using sequences near breakpoints (200 bp). **c** The ROCs of the bi-LSTM model for different single-cell datasets (validation data). The AUCs are also shown. **d** The PR curves and their AUPRs. **e** The densities of the technical artifact score of gene fusions in the PCAWG study by the bi-LSTM models retrained using six different datasets. **f** The densities of the predicted probabilities of chimeric reads. The models are retrained using different datasets. Source data are provided as a Source Data file.

**scFusion detects fusions with low false discoveries in a spike-in dataset.** To test whether scFusion can detect fusions in real scRNA-seq data, we introduced 27 known gene fusions to 27 single cells before performing single-cell sequencing[8,9] ("Methods" and Supplementary Data 7). All 27 fusions are from the PCAWG gene fusion study[35,36] and include well-known fusions such as *EML4-ALK*, *ROS1-GOPC*, and *TMPRSS2-ERG*. In total, we obtained scRNA-seq data from 729 single cells (~30 supporting reads for each fusion in each cell). scFusion reported all 27 spike-ins as well as 24 other fusions (Supplementary Data 8). All bulk methods except EricScript also detected all the spike-ins (Fig. 5a and Supplementary Data 9–12), but they reported a large number of fusions (310–9044, Fig. 5b), indicating their potentially high FDRs. We also performed bulk RNA-seq for the spike-in samples. 80% (41) of the fusions detected by scFusion were the spike-in fusions or had supporting chimeric reads in the bulk data, in contrast to the much lower percentage (0.5–17.7%) for the bulk methods (Fig. 5c), again indicating that the FDR of scFusion is much lower than the bulk methods.

**scFusion demonstrates high sensitivity in detecting marker fusions in a T cell dataset.** Another dataset we tested consists of 2355 T cells selected from ~7000 immune cells in patients with liver cancer[23]. The T cells of this population are non-malignant cells. Thus, other than the V(D)J recombinations of TCR genes,

gene fusions should be very rare in these cells. scFusion identified eight gene fusions, much fewer than bulk methods (Fig. 6a, Supplementary Data 13–17). Among the fusions detected by scFusion, six involve TCR genes, and four (50%) are V(D)J recombinations, indicating that many of these candidates are likely true positives. In comparison, only one (1.8%) of STAR-Fusion candidates are V(D)J recombinations and no candidates of other bulk methods are V(D)J recombinations (Fig. 6b). The two most frequent V(D)J recombinations identified by scFusion are *TRAJ33-TRAV1-2* and *TRAJ12-TRAV1-2*, with 126 and 20 supporting cells, respectively. Mucosal-associated invariant T (MAIT) cells are known to express the invariant *TRAJ33-TRAV1-2* and *TRAJ12-TRAV1-2* TCR α-chain[23,39]. Thus, the single cells with one of these fusions are likely MAIT cells. *SLC4A10*, a marker gene of MAIT cells[40], was expressed only in a cluster of T cells and many in this cluster had the *TRAJ33-TRAV1-2* or *TRAJ12-TRAV1-2* fusion (Fig. 6c, d). Differential expression analysis between the cells with and without these recombinations identifies 70 upregulated genes in the cells with the recombinations, among which *TRAJ33*, *TRAV1-2*, and *SLC4A10* are the top three most significantly upregulated genes (Supplementary Fig. 6a, b). Further, the *SLC4A10* expression is significantly associated with the fusion (Fisher's exact test, P-value < $10^{-16}$; Table 1).

These results indicate that the *TRAJ33-TRAV1-2* and *TRAJ12-TRAV1-2* fusions identified by scFusion are bona fide fusions and

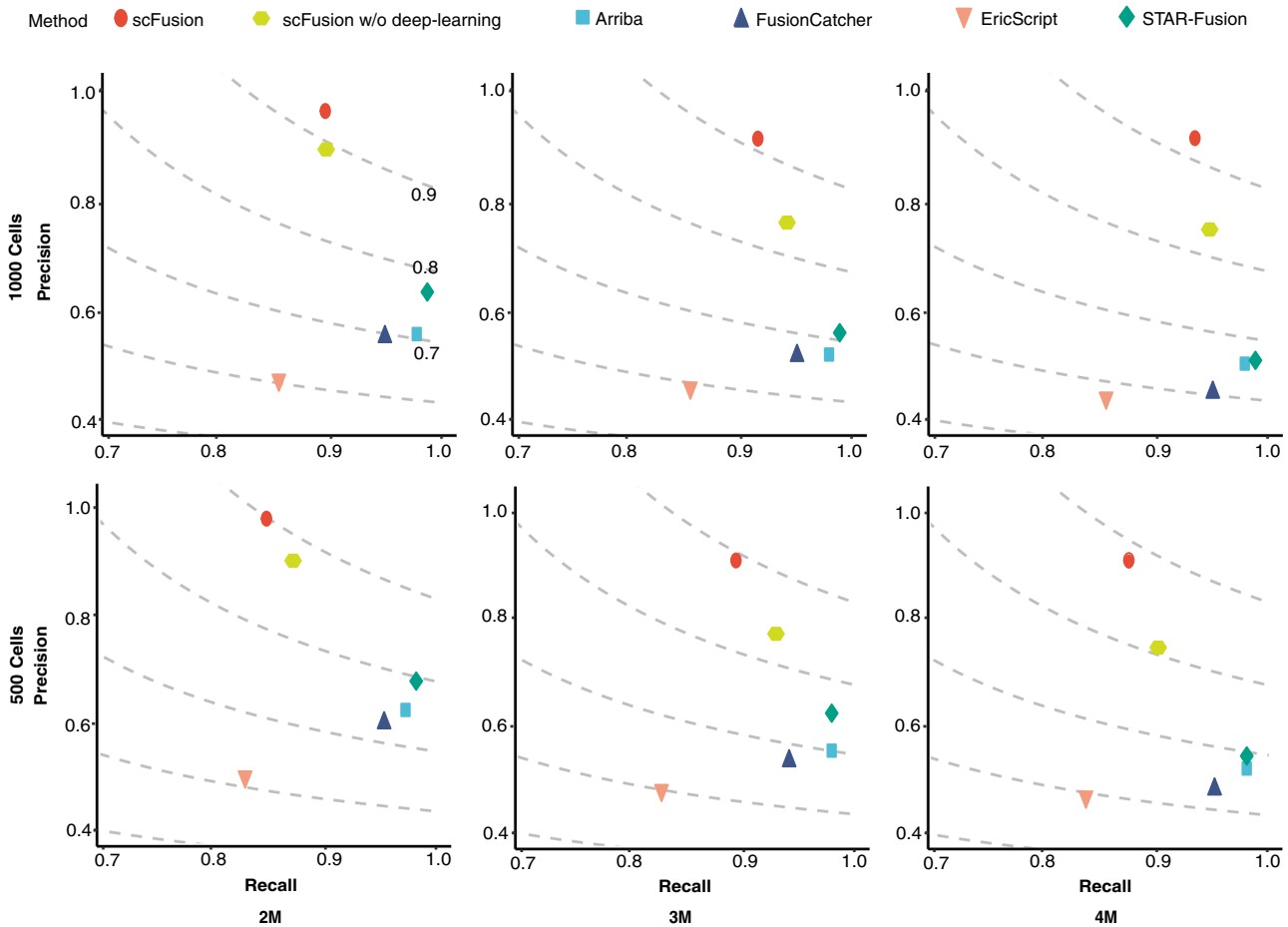

**Fig. 3 The precisions and recalls of scFusion, scFusion without the deep-learning model, and four bulk methods in six different simulation setups.** The figures in the two rows correspond to simulations with 1000 cells and 500 cells, and the figures in the three columns correspond to simulations with 2 million, 3 million, and 4 million reads in each data. The dots in the figures are the means of precisions and recalls of ten simulations in each setup. The dashed lines are the contour lines with constant F-scores (F-scores are marked in the top-left figure).

the cells with either fusion are MAIT cells. MAIT cells are an important component of the immune system. However, until recently, the detection of MAIT cells still relied on the combinations of cell markers and it is unclear how well these markers define the MAIT cells in different tissues or diseases[41]. scFusion provides an alternative way to sensitively define the MAIT cells. STAR-Fusion was the only bulk method that identified the *TRAJ33-TRAV1-2* fusion (62 cells) (Fig. 6e), and no bulk method reported any cells having the *TRAJ12-TRAV1-2* fusion (Fig. 6f), indicating that scFusion is more powerful in detecting shared fusions among single cells.

**Sensitive fusion detection by scFusion provides mechanistic insights in a multiple myeloma dataset.** We applied scFusion to a multiple myeloma (MM) dataset consisting of 597 single cells from 15 patients[33]. MM is a cancer of plasma cells. Approximately half of the myelomas have immunoglobulin heavy (*IgH*) chain translocations and around 10% of the myelomas have translocations involving immunoglobulin lambda (*IgL*) light chain locus[42,43]. In this dataset, scFusion identified 38 fusions while the bulk methods identified many more (41 to 1492, Fig. 7a, Supplementary Data 18-22). Around 94.7% (36) of scFusion candidates involve immunoglobulin genes (including 33 recombinations of immunoglobin genes), much higher than that for bulk methods. (Fig. 7b). scFusion successfully identified the recurrent *IgH-WHSC1* fusion in MM with two different breakpoints within *WHSC1*, positions 1902353 and

1905943 on chromosome 4 (Supplementary Data 18). The *IgH-WHSC1* fusions are in-frame fusions of *WHSC1*. All 52 cells with the breakpoint at 1902353 are from the patient SMM0. Similarly, all 20 cells with the other breakpoint are from the patient RRMM2 (Fig. 7c).

The expression of *WHSC1* is significantly higher in cells with the *WHSC1* fusions than in other cells (Fig. 7d). Interestingly, in cells with the *WHSC1* fusions, the sequencing coverage of *WHSC1* sharply increases downstream of the breakpoints, but the sequencing coverage of *WHSC1* in other cells is largely kept constant, indicating that the fusions probably lead to the overexpression of *WHSC1* (Fig. 7e and Supplementary Fig. 6c). *WHSC1*, also known as *NSD2* and a known oncogene[44], is one of the most commonly-fused partners with IgH in multiple myeloma[45–47]. Overexpression of *WHSC1* drives the chromatin change in an H3K36me2-dependent manner[48]. Differential expression analysis found 115 upregulated genes and 12 downregulated genes in the cells with the *IgH-WHSC1* fusions (q-value < 0.05 and log2 fold-change < −0.5 or >0.5) (Supplementary Fig. 6d and Supplementary Data 23). The upregulated genes include genes known to be co-expressed with *WHSC1* such as *MAL*[49] and *SCARNA22*[50]. The downregulated genes include known oncogenes in MM such as *CCND1* and *FRZB*. In fact, *CCND1* and *FRZB* tend to only express in cells without the *IgH-WHSC1* fusions (Supplementary Fig. 6e, f).

The bulk methods also detected the *IgH-WHSC1* fusions, but they reported fewer number of cells with the fusions than

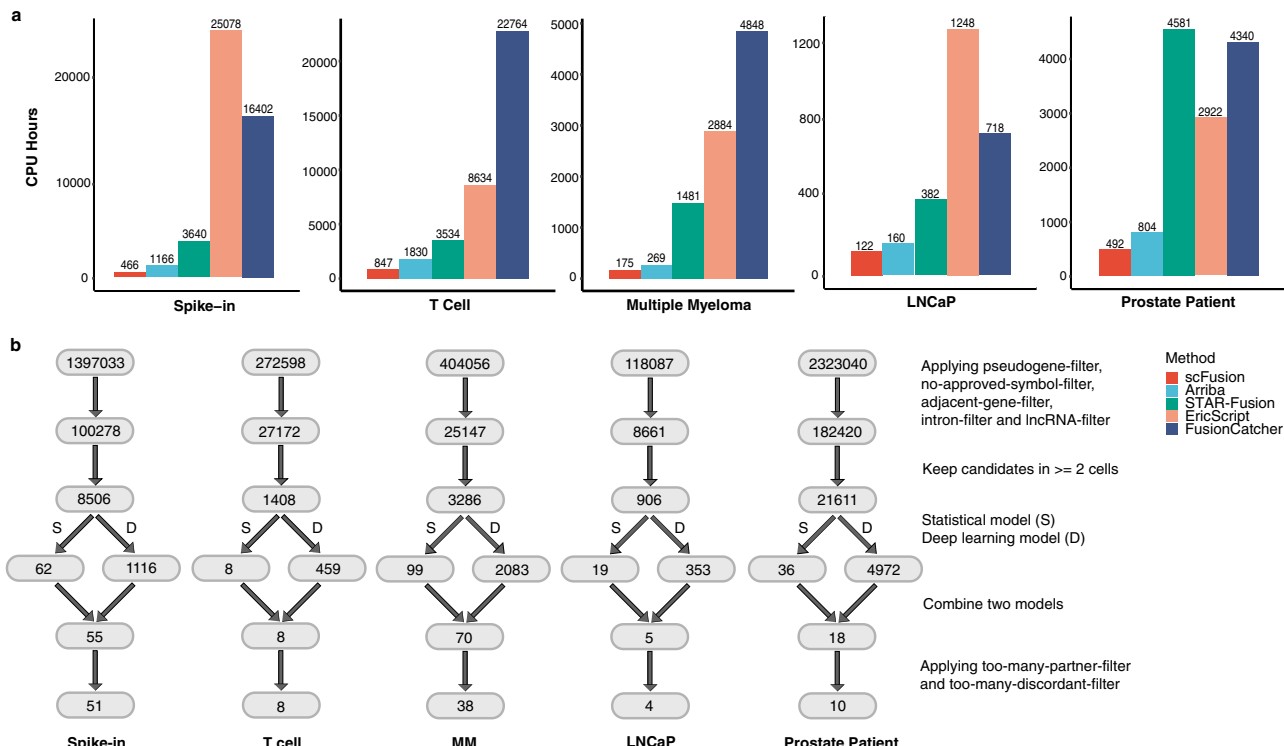

**Fig. 4 The computational time and effects of scFusion's filters. a** The computational time of the five methods for fusion detection in five scRNA-seq data. The *y*-axis is the total CPU hours of each method. **b** Number of fusion candidates after each filter. The top is the total number of fusion candidates, and the second layer is the number of fusion candidates after excluding candidates involving pseudogenes, lncRNAs, genes without an approved symbol, and in intronic regions (they are called pseudogene-, lncRNA-, no-approved-symbol-, and intron- filters). The third layer is the number of candidates supported by at least 2 cells. The fourth layer shows the numbers of fusion candidates remained after the filtering by the statistical model and the deep-learning model, respectively. The fifth layer is the number of candidates that pass both models. The last number indicates the number of output fusions after removing candidates with a partner gene involving in more than five fusion candidates and candidates whose number of supporting discordant reads is ten times more than their supporting split-mapped reads (too-many-partner- and too-many-discordant- filters).

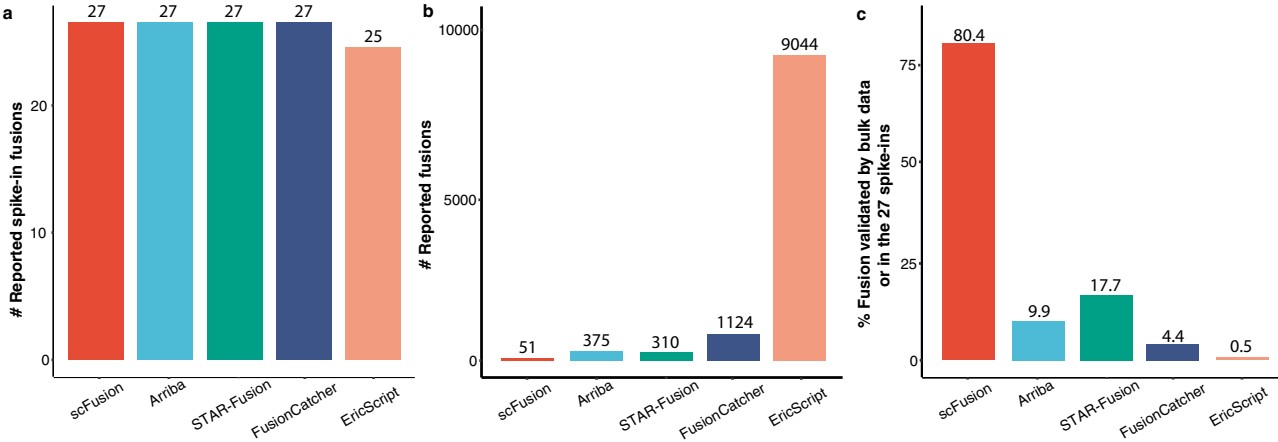

**Fig. 5 The performance of five methods on spike-in data. a** The numbers of reported spike-in fusions. **b** The numbers of reported fusions. **c** The proportions of fusions having bulk supporting chimeric reads or in the 27 spike-ins.

scFusion (Fig. 7f) and potentially many more false positives (Fig. 7a, b). The large number of false positives given by the bulk methods makes it very difficult for downstream analysis, thus limiting their application in single-cell analysis.

**scFusion detects known fusions in prostate cancer datasets.** Finally, we applied scFusion to two prostate datasets, from the cancer cell line LNCaP with 288 cells (LNCaP data)[31] and from 14 prostate cancer patients (prostate patient data)[38]. scFusion

detected 4 fusions in the LNCaP data whereas the bulk methods detected 14–288 fusions (Supplementary Fig. 7a and Supplementary Data 24–28). We compared the detected fusions with the fusions of the LNCaP cell line listed in the Cancer Cell Line Encyclopedia (CCLE) database. All of the four fusions reported by scFusion are in the CCLE fusion database and only 2.4–42.9% by the bulk methods are in the database, again indicating the high precision of scFusion. It is known that the LNCaP cell line does not have the well-known *TMPRSS2-ERG* fusion in prostate

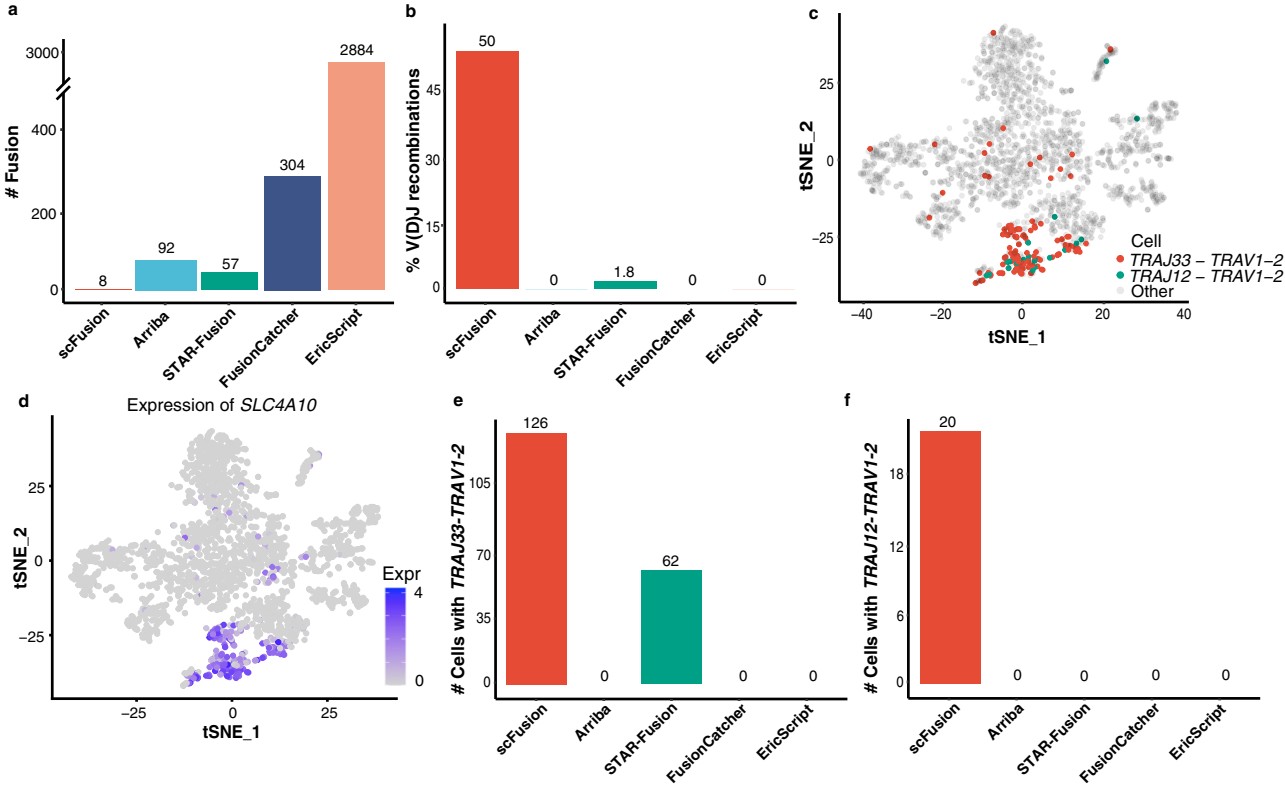

**Fig. 6 The T cell scRNA-seq data. a** The number of detected gene fusions by the five methods. **b** The percentages of V(D)J recombinations in fusions detected by the five methods. **c** The expressions of *SLC4A10* shown in the tSNE plot of all T cells. **d** The cells with *TRAJ33-TRAV1-2* and *TRAJ12-TRAV1-2* colored in the tSNE plot. **e**, **f** The barplots of numbers of cells with the *TRAJ33-TRAV1-2* (**e**) and *TRAJ12-TRAV1-2* (**f**) recombinations by different algorithms. Source data are provided as a Source Data file.

**Table 1 The contingency table between the *TRAJ33-TRAV1-2/TRAJ12-TRAV1-2* fusion and the expression of *SLC4A10*.**

|  | TRAJ33-TRAV1-2/TRAJ12-TRAV12 | Other |
|---|---|---|
| High expression | 108 | 90 |
| Low expression | 40 | 2,117 |

The high expression indicates the cells whose *SLC4A10* expression is greater than 1 and the low expression are other cells. Fisher's exact test (two-sided) gives a *p*-value smaller than $10^{-16}$.

cancers[51]. Consistent with this fact, no method detected the fusion.

The prostate patient dataset contains more single cells (922 epithelium cells), but the read length is only 38 bp, making fusion detection more difficult. scFusion only reported 10 fusions including *TMPRSS2-ERG* in 27 cells. The bulk methods reported 23–113 fusions including *TMPRSS2-ERG* in 17–26 cells (Supplementary Fig. 7b, c, and Supplementary Data 29–33). The cells reported to harbor the *TMPRSS2-ERG* fusions by different algorithms are largely consistent with scFusion (Supplementary Fig. 7d). Differential expression analysis shows that *ERG* is highly expressed in cells with the *TMPRSS2-ERG* fusion (Supplementary Fig. 7e), consistent with the previous research that *TMPRSS2-ERG* could lead to the overexpression of *ERG*[52]. Patient 1115655 had the largest number of cells (21) with the *TMPRSS2-ERG* fusion. scRNA-seq data of this patient before and after the enzalutamide (an androgen receptor inhibitor) treatment are available. Before the enzalutamide treatment, 16.7% (18/108) of cells contain the fusion, a much higher fraction than in cells after the treatment (3.7% or 3/81), consistent with the previous

observation that the *TMPRSS2-ERG* fusion can confer efficacy of enzalutamide[53] (Table 2). Differential expression analysis also shows that *ERG* is significantly downregulated in the cells after the treatment (Supplementary Fig. 7f).

## Discussion

A major challenge in fusion detection in single cells is the large number of false positives obtained when conventional methods are applied. To address this challenge, we introduced a statistical testing procedure to control the FDR. This procedure assumes that most fusion signals originate from technical noise and that true fusions generally have stronger signals than false positives. To remove systematic artifacts that may occur recurrently in multiple single cells (e.g., due to mis-priming during PCR amplification[37]), we developed a deep-learning model to learn the unknown technical artifacts and to filter the false positives generated by these artifacts. Another important difference compared to existing bulk methods is that scFusion takes advantage of the fact that multiple cells from a sample should contain the same fusion. Rather than considering each sample independently as bulk methods do, scFusion performs joint analysis of related cells, thus substantially increasing the detection power.

Although we have attempted to maximize its detection power, scFusion has reduced sensitivity for fusions with low expression (as shown in the simulation), just as is the case for all RNA-based methods. The detection power is also limited for rare fusions in highly heterogeneous tumor samples. These inherent limitations can be overcome only by sequencing more cells and/or sequencing each cell deeper.

One important application of scFusion, as illustrated by our detection of MAIT cells in our T cell data, is that fusions can

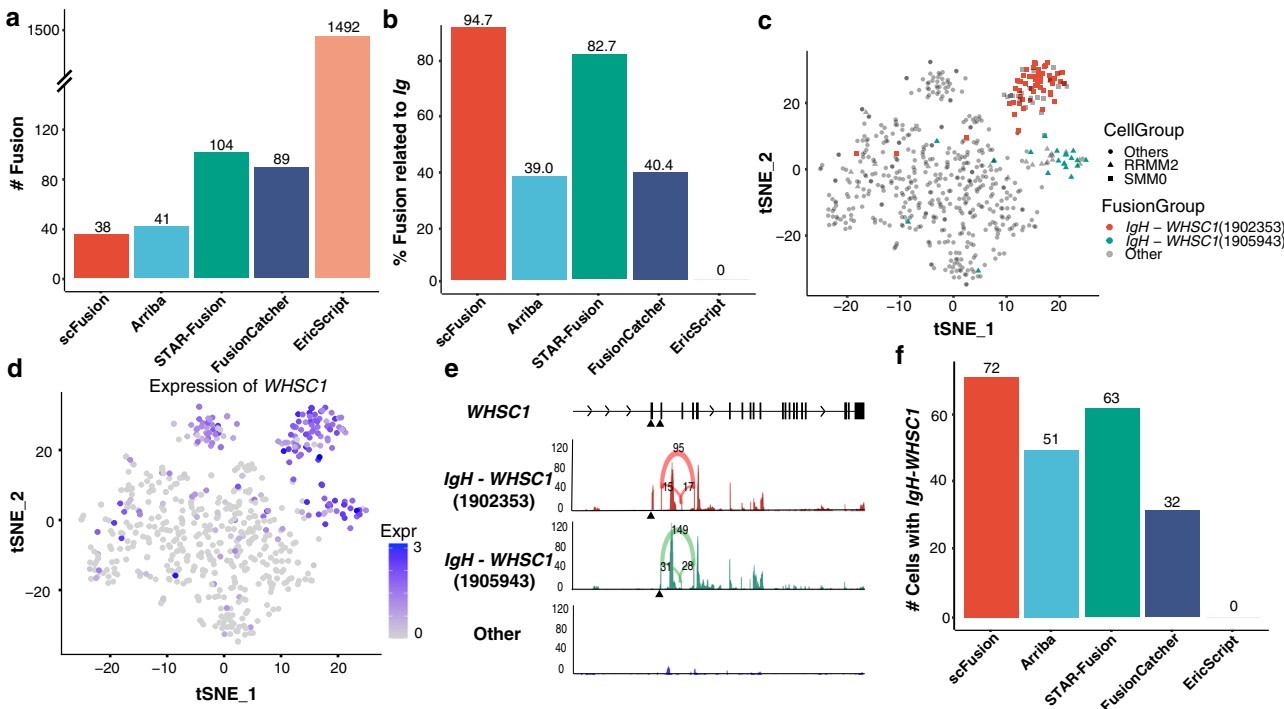

**Fig. 7 The MM scRNA-seq data. a** The number of detected gene fusions by the five methods. **b** The percentage of IgH-related fusions in fusions detected by the five methods. **c** The tSNE plot of all MM single cells. The cells with two *IgH-WHSC1* fusions are colored in the plot. The cells from patient RMM2 and SMM0 are marked by triangle and rectangle, respectively. **d** The expression of *WHSC1* shown in the tSNE plot. **e** The mean read depth of *WHSC1* at different locations for the cells with the two *IgH-WHSC1* fusions and the cells without the fusions. The black triangles indicate the breakpoints of the two fusions. The supporting number of splicing junctions are also shown in the plot (the numbers above the arcs). The read depth of a single cell at a location is calculated as the number of reads covering the location per million. The mean read depth is the average depth of all cells in a group. **f** The barplots of numbers of cells with the *IgH-WHSC1* fusions by different algorithms. Source data are provided as a Source Data file.

**Table 2 The contingency table between the *TMPRSS2-ERG* fusion and the enzalutamide treatment.**

|                  | With the fusion | Without the fusion |
| ---------------- | --------------- | ------------------ |
| Before treatment | 18              | 90                 |
| After treatment  | 3               | 78                 |

The Fisher's exact test (two-sided) gives a *p*-value of 0.0047.

serve to identify a subpopulation of cells in a given sample. Cells are typically clustered based on the expression level of a known marker gene or based on the overall similarity of the transcriptional profiles. However, choosing the appropriate thresholds to define distinct populations is a difficult issue. The presence of fusion transcripts would be one way of distinguishing a subpopulation, thus aiding in the cell type annotation task. Identifying such subpopulations may prove invaluable in studying drug-resistant or drug-sensitive subclones in a tumor.

In the real data analysis, we found that scFusion generally detected much fewer fusions compared with the bulk methods. Among the fusions detected by the bulk methods, many were very likely to be technical artifacts (Supplementary Fig. 8). For example, the *GADD45G-HSPH1* fusion was identified by all four bulk methods, but it contained a poly-A subsequence and its artifact score was almost 1. When deep-learning model was turned off, the precisions of scFusion reduced (Fig. 3, Supplementary Fig. 4, 9, and 10). We also noticed that many potential artifacts were also filtered by the ad hoc filters (Supplementary Fig. 11).

MAIT cells in humans are known to express three TCR α-chains, of which the most abundant is *TRAJ33-TRAV1-2*[41]. scFusion identified two of the three and found that *TRAJ33-TRAV1-2* is about 6 times more abundant than *TRAJ12-TRAV1-2*. The *TRAJ20-TRAV1-2* was not identified possibly because it was rare in MAIT cells. Gene differential expression analysis discovered 15 genes significantly differentially expressed between *TRAJ33-TRAV1-2* and *TRAJ12-TRAV1-2* cells (Supplementary Data 34). Interestingly, *ALPK1* and *TIFA* are highly expressed in the cells with *TRAJ12-TRAV1-2* (Supplementary Fig. 12). Recent studies[54] show that *ALPK1-TIFA* axis is a core innate immune pathway against pathogens such as *Helicobacter pylori*, implying that MAIT cells expressing different TCR α-chains might have different roles in the immune system. In the MM data, the *IgH-WHSC1* fusions are the most significant immunoglobulin-related fusions reported by scFusion, leading to the overexpression of the oncogene *WHSC1*[55]. *WHSC1* is highly expressed in single cells from three patients (SMM0, RRMM1, and RRMM2). scFusion did not find *WHSC1* fusions in single cells from RRMM2. It is likely that the expression of WHSC1 was activated in RRMM2 by mechanisms other than *WHSC1* fusions or that the fusion was missed by scFusion.

The scRNA-seq technology continues to improve, both in terms of the number of genes captured and the evenness of coverage across each transcript. As its data quality approaches that of bulk RNA-seq data, it will enable a more comprehensive profiling of fusion transcripts. We expect that scFusion and the statistical/machine learning framework introduced therein will find useful applications in future single-cell studies.

## Methods

**Statistical Model**. Suppose that there are $n$ cells and $N$ fusion candidates. Let $y_{ij}$ be the number of split-mapped reads supporting a fusion candidate $i$ in a cell $j$. The fusion candidate $i$ can either be a true fusion or from the background noises. If the distribution of the background noises is known, we can perform a statistical test to test if the observed numbers of supporting reads $y_{ij}$'s ($j = 1, \dots, n$) fit well to the distribution of the background noise. Candidates with significant p-values would be more likely to be true fusions. However, we do not know the distribution of the background noises and hence we need to first estimate this distribution. scRNA-seq data are count data and are often modeled by Poisson, negative binomial (NB) or zero-inflated negative binomial (ZINB) distributions[24–27]. The ZINB distribution is the most generalized version of these distributions. Since the data matrix of supporting chimeric reads has an excessive number of zeros (>95%), we assume that the distribution of the background noises is a ZINB distribution[25]. Estimating the distribution can be achieved by estimating the parameters in the ZINB distribution. Since we observed that the number of supporting chimeric reads depended on covariates such as the local GC content and expressions of partner genes (Fig. 2a, b), we establish a regression model to link the parameters in the ZINB distribution and the covariates. The parameters of the regression model can be estimated by the maximum likelihood estimation. Details of the model are given below.

Denote $Y_{ij}$ the random variable corresponding to $y_{ij}$. Since the candidate fusions have very few true fusions, we can safely use all $y_{ij}$ to estimate the distribution of the background noises. Assume that $Y_{ij}$ follows the ZINB distribution[25]. The ZINB distribution has three parameters. One parameter is $p_i$, the probability that the cell $j$ has no split-mapped read supporting the fusion candidate $i$ or $p_i = P\left(Y_{ij} = 0\right)$. The other two parameters of the ZINB are the mean $\mu_{ij}$ and the overdispersion parameter $\lambda$ of the negative binomial (NB) distribution. Thus, we have $Y_{ij} \sim \text{ZINB}(p_i, \mu_{ij}, \lambda)$. Conditional on $Y_{ij} > 0$, $Y_{ij}$ follows a zero-truncated NB distribution. We assume that the ZINB distribution depends on the expression levels of the partner genes and the GC content[56]. Let $e_{1ij} < e_{2ij}$ be the expressions of the two partner genes corresponding to $Y_{ij}$ and $t_i$ be the GC content of the exonic sequence near the junction of the candidate gene (200 bp). Further, define $\bar{e}_{1i} < \bar{e}_{2i}$ as the mean expression of the two partner genes of the candidate fusion $i$ across all single cells. The gene expression is largely linearly dependent on the supporting read number (Fig. 2a) and the regression functions against the gene expression is set as linear. The GC-content dependency is nonlinear (Fig. 2b) and is represented by spline functions. Thus, we consider the following regression model

$$\text{logit}(p_i) = \beta_{10} + f(t_i) + \beta_{11}\bar{e}_{1i} + \beta_{12}\bar{e}_{2i}, \tag{1}$$

$$\log\left(\mu_{ij}\right) = \beta_{20} + g(t_i) + \beta_{21}e_{1ij} + \beta_{22}e_{2ij}, \tag{2}$$

where $f$ and $g$ are unknown functions, and the $\boldsymbol{\beta}_1 = (\beta_{10}, \beta_{11}, \beta_{12})$ and $\boldsymbol{\beta}_2 = (\beta_{20}, \beta_{21}, \beta_{22})$ are unknown parameters. We represent $f$ and $g$ using spline functions, $f(x) = \sum_{k=1}^{K}\beta_{kf}B_k(x)$ and $g(x) = \sum_{k=1}^{K}\beta_{kg}B_k(x)$, where $K$ is the number of spline base functions (by default $K$ is set as 5). Denote $\boldsymbol{\theta} = (\boldsymbol{\beta}_1, \boldsymbol{\beta}_2, \boldsymbol{\beta}_f, \boldsymbol{\beta}_g)$ with $\boldsymbol{\beta}_f = (\beta_{1f}, \cdots \beta_{Kf})$ and $\boldsymbol{\beta}_g = (\beta_{1g}, \cdots \beta_{Kg})$. We estimate the unknown parameters $\boldsymbol{\theta}$ by maximizing the following likelihood (MLE),

$$l(\beta_1, \beta_2, \lambda) = \prod_{i=1}^{N} l_i\left(\boldsymbol{\theta} \bigg| \sum_{j=1}^{n} 1_{y_{ij}>0} \geq 2\right), \tag{3}$$

$$l_i\left(\boldsymbol{\theta} \bigg| \sum_{j=1}^{n} 1_{y_{ij}>0} \geq 2\right) = \frac{p_i^{\sum_{j=1}^{n} 1_{y_{ij}=0}} \prod_{j:y_{ij}>0} \frac{\phi(y_{ij};\mu_{ij},\lambda)}{1-\phi(0;\mu_{ij},\lambda)}}{1 - p_i^n - n\left(1 - p_i\right)p_i^{n-1}}, \tag{4}$$

where $N$ is the number of fusion candidates supported by at least 2 cells, $n$ is the number of cells, $\phi$ is the probability density function of NB distribution with mean $\mu_{ij}$ and overdispersion $\lambda$. Note that to reduce the computational burden, we only consider the candidates with at least two supporting cells and thus the likelihood $l_i$ for the candidate $i$ is a conditional probability.

**Statistical Test for Significant Fusions**. Plugging-in the MLE estimates of the regression parameters, we can estimate the background noise distribution for each fusion candidate. With the distribution estimate, we can test if the observed number of supporting chimeric reads for the fusion candidate is likely to be sampled from the background distribution. If the background distribution is unlikely to generate an observation that is larger than the observed supporting reads number, or in other words, if we obtain a very small p-value, we reject the null hypothesis that the fusion candidate is from the background noise and retain the fusion as a true fusion candidate. Details of the testing procedure are given below.

Let $\hat{\boldsymbol{\theta}} = (\hat{\boldsymbol{\beta}}_1, \hat{\boldsymbol{\beta}}_2, \hat{\boldsymbol{\beta}}_f, \hat{\boldsymbol{\beta}}_g)$ and $\hat{\lambda}$ be the MLE of the parameters $\boldsymbol{\theta}$ and $\lambda$, respectively. Plugging-in $\hat{\boldsymbol{\theta}}$ to the formula (1) and (2), we can get the parameter estimates $\hat{p}_i, \hat{\mu}_{ij}$ and $\hat{\lambda}$ of the ZINB distribution and can estimate the ZINB distribution by $\text{ZINB}(\hat{p}_i, \hat{\mu}_{ij}, \hat{\lambda})$. We use a resampling scheme to obtain a p-value for each fusion candidate. Specifically, for

each fusion candidate $i$, we sample $\widetilde{Y}_{ij}^{(b)}$ from $\text{ZINB}(\hat{p}_i, \hat{\mu}_{ij}, \hat{\lambda})$ for each cell $j$ and obtain the sum $\widetilde{S}_i^{(b)} = \sum_{j=1}^{n}\widetilde{Y}_{ij}^{(b)}$ ($b = 1, \cdots, B$). Denote $\widetilde{\nu}_i$ and $\widetilde{\sigma}_i^2$ be the sample mean and variance of $\widetilde{S}_i^{(b)}$, respectively. The distribution of $\widetilde{S}_i^{(b)}$ can be approximated by a normal distribution with the mean and variance approximately $\widetilde{\nu}_i$ and $\widetilde{\sigma}_i^2$, respectively. Let $S_i = \sum_{j=1}^{n} Y_{ij}$. The p-value for the fusion candidate $i$ is set as $p_i = 1 - \Phi((S_i - \widetilde{\nu}_i)/\widetilde{\sigma}_i)$, where $\Phi$ is the cumulative distribution of the standard normal distribution. We set $B = 1000$ in all analyses.

To determine a p-value cutoff to control the FDR, we split our candidate fusion set into two subsets. The first subset is a high-quality subset that more likely contains the true gene fusions and the second subset is the remaining candidates. Suppose that $n_1$ and $n_2$ as the total number of candidates in the first and the second subset, respectively. Given a p-value cutoff $c$, let $m_1(c)$ and $m_2(c)$ be the number of candidates in the two subsets with p-values less $c$, respectively. Since the second subset contains much fewer true positives than the first subset, $(n_1 + n_2)\frac{m_2(c)}{n_2}$ can be used as an estimate of total number of false discoveries at the p-value cutoff $c$ and thus the FDR is roughly $\text{fdr}(c) = \frac{m_2(c)}{n_2}\frac{n_1+n_2}{m_1(c)+m_2(c)}$. We choose $c$ such that $\text{fdr}(c) \leq \alpha$ for a given $\alpha$ (usually 0.05). In general, we believe that candidates with more supporting cells are more likely to be true. Hence, we choose the first subset as the fusion candidates with at least 1% of the total cells having the fusion and with at least $s$ supporting reads per cell on average, where $s$ is taken as 1.25 in all simulation and real data analysis.

**The bi-LSTM**. Each single cell usually has over 10,000 chimeric reads (Supplementary Fig. 3c) and the vast majority of them are obviously not signals of gene fusions but technical artifacts. We set the negative training data as the subsequences from chimeric reads, representing the technical artifacts. An ideal positive training data would be chimeric reads from true fusions. Unfortunately, we do not know which chimeric reads are from true fusions. Even if we knew some, the number of such reads would be too small to train the bi-LSTM network. To overcome this challenge, a proxy task strategy is applied: we set the positive training data as sequences generated by concatenating random pairs of short reads. Since the sequences are randomly generated, they should not contain features of technical chimeric reads and hence characteristics of technical artifacts could be learned by comparing with these random sequences. More specifically, the negative training data of the bi-LSTM is chosen as the subsequences (60 bp) of chimeric reads covering the junction positions and the junction positions are required to be at 15-45 bp of the subsequences. The positive training data is the similar 60 bp subsequences of randomly concatenated short read pairs.

In the bi-LSTM (Supplementary Fig. 1b), the DNA sequences, with 60 nucleotides and one fusion site, are given as the input to the embedding layer. The four types of nucleotide (i.e., A, T, G, and C) and the fusion site are represented by five different 5-dimensional feature vectors. The output of the embedding layer is passed on to three sequence-to-sequence bi-LSTM layers (with 32, 64, and 128 bi-LSTM units, respectively) and further to a sequence-to-one bi-LSTM layer for sequential feature extraction. The extracted features are fed to two fully connected layers and finally to a softmax layer to produce the softmax probabilities of the read being classified to technical chimeric artifacts. In the training process, the binary cross-entropy is used as the loss function and model parameters are updated using the Adam optimizer[57]. The model is pre-trained for 200 epochs with the batch size set to 500.

Using this bi-LSTM model, scFusion gives each fusion candidate a technical artifact score. By default, fusions with scores greater than 0.75 are filtered. The training step is computationally expensive. To expedite the training, scFusion provides a pre-trained model that can be used as the initial value to train bi-LSTM models for new datasets. The retraining is trained for 30 epochs with the pre-trained model as the initial value. Note that if we directly used the pre-trained model, the median AUC and AUPR were 0.673 and 0.749 (Supplementary Fig. 2f, g), respectively. The convolutional neural network (CNN) is another popular neural network model used in many biological applications[58]. We also built a CNN model for comparison and found that the bi-LSTM generally performed better than the CNN (Supplementary Material, Supplementary Fig. 13). Hence, we use the bi-LSTM in all data analyses.

**Simulation setup**. The simulation data was generated using FusionSimulatorToolkit[20]. This toolkit can simulate RNA-seq data with gene fusions by learning from a real RNA-Seq dataset such as its expression, insert size, read length and mutation rate. We provided the immune cell scRNA-seq[23] data to FusionSimulatorToolkit to train a single model and provided this model to FusionSimulatorToolkit to generate RNA-seq data with 100 simulated fusions from PCAWG at various gene expression levels. For technical chimeric artifacts, we design a new method to add technical chimeric artifacts to the simulated data. Briefly speaking, we randomly sample transcript pairs to generate technical fusions between the sampled transcript pairs. The sampling probabilities of the transcript pairs depend on their sequence features to mimic the random annealing and mis-priming in PCR amplification[37]. After a technical fusion is introduced to a cell, we further assign a random expression to the technical fusion and add short reads from the fusion to the cell.

More specifically, given any two transcripts $T_1$ and $T_2$ (or their reverse complements) and their potential breakpoints $b_1, b_2$, we define a mis-priming

likelihood

$$A(T_1, b_1, T_2, b_2) = \frac{1}{1 + \exp(180 - 3\psi(T_1, b_1, T_2, b_2))}, \quad (5)$$

where $\psi(T_1, b_1, T_2, b_2)$ is the "binding energy" between $T_1$ and $T_2$ near the breakpoints $b_1, b_2$. Here, we define

$$\psi(T_1, b_1, T_2, b_2) = \sum_{i=1}^{6} \phi(S_{1i}, S_{2i}), \quad (6)$$

where $S_1 = S_{11} \cdots S_{16}$ ($S_2 = S_{21} \cdots S_{26}$) is the 6-mer subsequence of $T_1(T_2)$ near $b_1(b_2)$, $\phi(A, T) = \phi(T, A) = 12$, $\phi(G, C) = \phi(C, G) = 21$ and $\phi(N_1, N_2) = 0$ for all other nucleotide pairs ($N_1, N_2$). Then, using $A(T_1, b_1, T_2, b_2)$ as the weighting probability, we randomly generate a library of 2.2 million potential technical chimeric sequences. If $(T_1, b_1, T_2, b_2)$ is sampled, the corresponding chimeric sequence is chosen as $T_{11}T_{22}$, where $T_{11}$ is the subsequence of $T_1$ upstream of $b_1$ and $T_{22}$ is the complementary of the subsequence of $T_2$ downstream of $b_2$. Then, for each cell, we randomly select chimeric sequences from this library with a new sampling weight

$$W(T_1, b_1, T_2, b_2) = \frac{\psi(T_1, b_1, T_2, b_2)}{1 + \exp(195 - 3\psi(T_1, b_1, T_2, b_2))} \quad (7)$$

and generate chimeric short reads from the selected sequences. The expressions of the technical chimeric sequences are randomly sampled from 1 to 100. The total number of chimeric reads for each cell is set as around 1% of the total number of reads, at the similar level as real scRNA-seq data.

**Gene fusion spike-in and single-cell sequencing.** cDNA sequences of 27 fusion genes were synthesized and constructed into independent lentiviral vectors. Every lentiviral vector along with two auxiliary packaging plasmids was co-transfected into independent 293T cells. After 48 hours, the supernatant was collected from the 293T cells and filtered through a 0.45 uM membrane. The 27 different recombinant lentiviral particles containing the target fusion genes and the green fluorescent protein (GFP) reporter gene were collected. Then, the 27 recombinant lentivirus particles were infected into 293 T cells, respectively. 72 hours after infection, the medium was changed and the expression of GFP in cells was checked under a fluorescence microscope to determine if lentivirus infection was successful. After the infection, 27 cell cultures expressing different target fusion genes were collected. The collected cells were washed with 1x PBS and resuspended with 2 mL PBS. Single-cell sorting was performed on the BD Biosciences FACS-ARIA platform and the single GFP positive cells (27 cells from each of the 27 cell cultures) were screened into 96-well PCR plates, respectively, for the next step of single-cell RNA library construction. Poly(A)-transcripts of total RNA of single cells were reverse transcribed and amplified using the SMART-seq2 protocol. The amplified cDNA was tagmented by Nextera XT kit (Illumina) and libraries were sequenced by NovaSeq (Illumina). The 293 T cells were purchased from the National Infrastructure of Cell Line Resource (http://www.cellresource.cn/).

**scRNA-seq data analysis.** For scFusion, short reads are first aligned with STAR(v 2.7.4a) to the human reference genome (hg19). scFusion only considers short reads mapped to uniquely mappable positions in the exonic regions of the reference genome (uniquely mappable for 75 bp sequences). Then, the split-mapped reads are clustered. If the breakpoints of two split-mapped reads are no larger than 20 bp away from each other, the two split-mapped reads are clustered together. A fusion candidate's breakpoint is taken as the median of the breakpoint positions of its all supporting chimeric reads.

FusionCatcher(v1.10) was run on the default parameters. Arriba(v1.0.1), EricScript(v0.5.5b), and STAR-Fusion(v1.8.1) were all run on the default parameters except the minimum-supporting-reads parameter. We tuned the minimum-supporting-reads parameter for all bulk methods except FusionCatcher (since it does not have any tuning parameter) using the simulation data. We found that the bulk methods had the highest F-scores when the parameter was set as three. So we set the minimum-supporting-reads parameter as 3 for EricScript, STAR-Fusion and Arriba in all data analyses. The reference genome was chosen as hg19. For the bulk methods, we also applied the same ad hoc filters of scFusion to make different algorithms comparable. Specifically, we applied the pseudogene-, lncRNA-, no-approved-symbol-gene-, intron-, too-many-partner- and too-many-discordant- filters to the fusion candidates of bulk methods. lncRNA- and no-approved-symbol-gene- filters are optional, and Supplementary Fig. 14 shows the numbers of detected fusions in each dataset when these two filters are turned off.

**Gene expression analysis.** The expression matrix of T cell and the multiple myeloma dataset were downloaded from the Gene Expression Omnibus (GEO) (https://www.ncbi.nlm.nih.gov/geo/). First, we used Seurat (v 3.2.0)[59,60] to read the Transcripts Per Million (TPM) matrix. The expression was further normalized using the NormalizeData function in Seurat. The highly variable genes were identified using the function FindVariableGenes. Their expression was scaled and centered along each gene using ScaleData. Then we performed dimension reduction using principal component (PC) analysis. We selected the first 30 PCs for t-distributed stochastic neighbor embedding (tSNE), and tSNE plots were generated using Seurat. To identify differentially expressed genes, we used the function FindAllMarkers in Seurat with the Wilcoxon rank-sum test. Genes expressed in at least 10% cells within the cluster and with

the log fold-change more than 0.5 and the adjusted p-value smaller than 0.05 were considered as differentially expressed genes.

**Statistics and Reproducibility.** The linear correlation tests were performed by two-sided Student's t test. The significance of difference in the contingency tables was tested by Fisher's exact test (two-sided). The tests on differential expressed (DE) genes were performed by two-sided Wilcoxon rank-sum test adjusted by Benjamini-Hochberg procedure. A p-value or adjusted p-value lower than 0.05 was considered significant.

In this study, we spiked-in gene fusions to single cells and performed single-cell RNA sequencing using SMART-seq2 to evaluate the fusion detection algorithm. Simulation showed that sensitive detection of gene fusions need ~20 cells for each fusion and we chose to spike-in 27 cells for each fusion. We chose to spiked-in 27 gene fusions for stable evaluation of the algorithms. This gave us 729 cells of scRNA-seq data, largely similar to the cell numbers in studies using full-length scRNA-seq techonologies[23,33]. No statistical method was used to predetermine sample size. No data were excluded from the analyses. The experiments were not randomized. The Investigators were not blinded to allocation during experiments and outcome assessment.

**Reporting summary.** Further information on research design is available in the Nature Research Reporting Summary linked to this article.

## Data availability

Raw data of the spike-in data generated in this study have been deposited in the Genome Sequence Archive (GSA) of BIG Data Center, Beijing Institute of Genomics (BIG), Chinese Academy of Sciences, with an accession code HRA001199. The scRNA-seq data used in this study are available in Gene Expression Omnibus (GEO) under accession code GSE81812, GSE99795, GSE110499, GSE118900, GSE127298, GSE140228. Raw sequence data of the prostate patient data used in this study are available in dbGaP under accession codephs001988.v1.p1 (https://www.ncbi.nlm.nih.gov/projects/gap/cgi-bin/study.cgi?study_id=phs001988.v1.p1). Source data are provided with this paper.

## Code availability

scFusion is available at GitHub (https://github.com/XiDsLab/scFusion)[61].

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

## Acknowledgements

This work was supported by the National Natural Science Foundation of China [No. 11971039 to R.X.], the National Key Basic Research Program of China [2020YFE0204000 to R.X.], and Sino-Russian Mathematics Center. Part of the analysis was performed on the High Performance Computing Platform of the Center for Life Sciences (Peking University).

## Author contributions

R.X. conceived and supervised the study. Z.J. developed the statistical model. W.H., Z.J., and X.W. developed the deep-learning model. J.L. and J.D. performed the single-cell experiment. Z.J., R.X., and N.S. performed data analysis. R.X., Z.J., N.S., W.H., and X.W. wrote the manuscript, with important feedback from P.P.

## Competing interests

The authors declare the following competing interests: Ruibin Xi holds the stock of GeneX Health Co.Ltd. A patent application about single-cell gene fusion detection is submitted. Applicant: Peking University. Inventors: Ruibin Xi, Zijie Jin. Application number: 202011451710.8. Status of the application: pending. The algorithm developed in this manuscript is covered in the patent application. For all other authors, no competing interests exist.
