## [Peer Review File · Nature Communications]

Reviewers' Comments:

Reviewer #1:

Remarks to the Author:

The authors describe a method for detecting fusion transcripts in single cell transcriptome data, which they implemented in a tool called 'scFusion'. This method involves first leveraging STAR to align scRNA-seq reads to the genome to identify chimeric reads, and then after identifying candidate fusion transcripts, advanced computational methods are leveraged for filtering likely artifacts. At the heart of scFusion are two methods for fusion filtering: a statistical method leveraging a zero-inflated negative binomial model to describe the background (null distribution) of fusion supporting reads as a function of fusion gene partner expression levels and GC content, and a separate neural network to filter chimeric read alignments based on breakpoint sequence composition to differentiate likely artifacts from randomly fused sequences. The authors apply scFusion and alternative bulk-rna focused methods to single cell transcriptomes in various contexts: simulated scRNA-seq, spiked-in scRNA-seq, and publicly available cancer-relevant single cell rna-seq data sets, and the authors argue that scFusion outperforms alternative methods in each scenario.

Fusion detection in single cell transcriptomes is an important and challenging problem, and the algorithmic developments underlying scFusion are well-intentioned, but I suspect that several experimental design considerations were misguided and deserve further consideration.

Major critiques:

The competing bulk rna-seq fusion detection methods were evaluated based on their default parameterizations, which would in most cases not be well suited to single cell transcriptomes. Most methods require at least two to three reads supporting a fusion for a prediction. For peak sensitivity with single cell data, it would be necessary to run the bulk methods with parameters tuned for improved sensitivity. It would appear that scFusion requires only one fusion read to support a prediction for an individual cell (based on the supplementary data provided), which may largely explain its improved sensitivity in certain contexts (eg. improved sensitivity for IgH-WHSC1 detection).

The way the authors score false positives and compute specificity is troubling. For simulated data, it is straightforward to identify false positives because the authors have control over which fusions are simulated. However, in the simulations performed by the author, the original chimeric reads from single cell experiments were added to the simulated data as a way of including artifact reads. The problem here is that there are very likely to be real fusions supported by those chimeric reads in addition to true artifacts, and fusions called based on those chimeric reads may in some cases be true fusions and not false positives. Those fusions being called that appear to be neighboring genes and potentially cis-spliced, or involving local rearrangements, are all least likely to derive from artifacts and could be reconsidered.

Rigorous benchmarking was performed in only certain limited contexts. From the simulations (Figure 3), it is clear that STAR-Fusion and Arriba have high sensitivity that surpasses that of scFusion, but both are penalized for having low precision, and I strongly suspect this is because of the addition of the chimeric read supplements from the real single cell data and certain of these putative false positives are likely to be true fusions present in the genuine single cell data. The authors should include experiments performed with simulated data that are not supplemented with real data to avoid this issue.

It is curious that the authors focused their primary findings based on T-cell receptor arrangements in immune cells and IGH fusions in multiple myeloma. There are specialized methods that focus on T-cell receptor configurations (such as TraCeR or TRUST) and one doesn't typically use a chimeric gene finder for this purpose. In regard to IGH-fusions, these tend to be especially challenging to identify in comparison to most fusion transcripts relevant to cancer (ie. most PCAWG fusions). The authors targeted six publicly available single cell RNA-seq data sets, yet only focus the neural network chimeric read filtering accuracy on all six. Little attention is given to other fusions more typical of fusion-finding efforts in transcriptomes. While one of the single cell data sets

corresponds to prostate cancer, no discussion was provided in the paper regarding how well the hallmark TMPRSS2-ERG fusion was identified by scFusion and alternative methods. A more comprehensive look at fusion-finding capabilities of scFusion in comparison to alternatives is needed.

For the simulations leveraging the Fusion Simulation Toolkit, from the methods description it isn't clear how the data were simulated for single cells, given that the toolkit was developed toward simulating bulk data. Was each cell targeted and modeled separately via the read simulator? Or was a single model created and run N times to generate N cells worth of scRNA-seq data? Ideally, it's the former.

Extended data figure 1 shows the impact of the filtering methods implemented in scFusion and would best be shown as a main figure instead of being restricted to the supplement. From this figure, it is clear that the statistical model is most responsible for the selection of candidate fusions, with only small contributions from the neural network towards filtering potential artifacts. This should be made clear in the main text of the manuscript.

Finally, it is curious that even after applying the advanced computational methods for filtering candidates, there continues to be additional 'two-many ...' filters (extended data figure 1), and what appears to be 'badgene' listing that assists in additional filtering (found in the code). It isn't clear to me what extent these additional 'badgene' filters contribute to overall fusion prediction capabilities, and this should be made clear in the manuscript as well.

Comments regarding the software:

I encourage the authors to create a docker image containing all needed software so it can be easily run as a single package, without requiring additional setup time on the part of the user.

The hg19 mappability file could be included with the software instead of requiring the user to download it separately from the ucsc track downloader.

Reviewer #2:

Remarks to the Author:

The manuscript entitled "Single cell gene fusion detection by scFusion", by Dr. Zije Jin and colleagues, describes scFusion, a software tool for detecting gene fusions in full-length single cell gene expression datasets. The tool uses a combination of statistical modeling and machine learning (ML) to identify true fusion events from a high level of data noise.

This work is of potentially high significance, as scFusion appears to be the only method to date designed for fusion detection in single-cell data, although the authors don't state this clearly. Based on the evaluations included in the study, the method has higher sensitivity and accuracy than other, bulk RNA sequencing based approaches. It also runs faster (although still consuming 100s of hours of CPU time per analysis). Despite these strengths, the manuscript has a number of weaknesses, as detailed below.

Biological significance:

- (1) The authors utilize published datasets, with the exception of a dataset generated for addressing a technical point, therefore the manuscript reports no biologically novel data.
- (2) Similarly, the published datasets are used for evaluating the method, without any essential new insight.

Presentation:

- (3) The presentation is often unclear, to the point where it simply prevents unambiguous understanding of the material. For example, I was simply not able to clearly interpret the

statement "Finally, scFusion filters out fusion candidates whose number of supporting discordant reads is ten times more than the supporting split-mapped reads and the candidates with a partner gene involving in more than five fusion candidates.". The clarity of the presentation must improve to make this manuscript accessible for the authors (not to mention the reviewers).

(4) Along similar lines, the description of the methods is restricted to mathematical details and is therefore inaccessible. I could gain very little insight into the inner workings of the algorithms.

Technical issues:

(5) There are some unjustified methods in generating the simulated datasets use in the tool evaluations. For example, the method of generating a positive training dataset by concatenating random pairs of short reads does not appear to faithfully represent true fusions. As another example, the authors mixed "real" fusion transcripts with their simulated background data, which is potentially problematic. Especially when using machine learning, it is entirely possible for the ML algorithm "learns" to identify such spike-ins based on extraneous differences between the simulated and the real sequencing reads, rather than recognizing the fusion. This effect could be potentially quantified by spiking in additional "real" data without fusions.

(6) The argument that scFusion works better than other methods in the T-cell dataset simply because it calls less fusions than the other methods, and arguing that these extra detected fusions must be false positives is circular.

(7) Related to the previous point, if all detected fusion candidates (in addition to the spike-ins) are indeed false positives, the scFusion false discovery rate (FDR) would be $10/(10+4)$, over 70% considering 10 false positives and 4 true positives. This is potentially too high for practical application.

Reviewer #3:

Remarks to the Author:

Within the manuscript "Single cell gene fusion detection by scFusion", the authors detail a new single cell sequencing computational tool that precisely detects gene fusions from RNA-seq data. They describe the development of scFusion, in addition to demonstrating the utility of the tool using a combination of simulated, spiked and publicly available cancer datasets.

In contrast to other approaches – which repurpose computational tools for bulk RNA-seq analysis – the development of scFusion represents the first analytical tool specifically designed to detect gene fusions from a single cell source. As such, scFusion is a welcome addition to the current methods available for detection and analysis of gene fusions in cancer patient samples.

However, there are a few key issues relating to the manuscript that require elaboration or amendment –

Methods comment:

1) "Gene fusion spike-in and single cell sequencing." Why did the authors approach the experiment in this way? Attempting to transfect cells with multiple constructs at the same time is a very random approach. Why did the authors not transfect separate 293T cell cultures independently with a single construct, pool successfully transfected cultures at known proportions and then use this as input for scRNA-seq? This would provide more accurate knowledge of the starting material and provide a blueprint upon which to design further assays to change the titration amounts of each transfected cell and precisely determine the lower detection limits of scFusion. With the current approach, the authors can only state in the Results section that "scFusion did not detect the spiked-in fusion CCDC6-RET, since the fusion occurred in only two cells and did not achieve the significance level.", which gives no indication of the true detection limit of the computational tool.

Results comments:

1) For the scFusion method, the authors state that, "Finally, scFusion filters out fusion candidates whose number of supporting discordant reads is ten times more than the supporting split-mapped reads and the candidates with a partner gene involving in more than five fusion candidates.". Could the authors detail how they defined / assigned these seemingly arbitrary numerical values?

2) For the simulated data, the authors state that, "The fusions missed by scFusion are mostly fusions with low expression levels (Extended Data Fig. 5)". Could the authors please comment on whether this could be an additional limitation for scFusion if it was applied to scRNA-seq data from a highly heterogeneous tumour tissue sample?

Discussion comment:

1) The authors conclude with, "Analysis of gene fusions at single cell level will provide unprecedented opportunities to study their roles in tumor development, tumor heterogeneity as well as tumor cell's responses to various pharmaceutical therapies.". However, the computational tool has only been applied in vivo to examples of multiple myeloma, pancreatic cancer (which does not harbour disease-associated gene fusions) and artificially spiked cell line datasets. Without further demonstrated application of scFusion to other blood cancer and solid tumour datasets, I find this statement somewhat overreaching.

General comment:

1) There are small spelling and grammatical errors throughout the submission that should be resolved.

Responses to Reviewers' Comments

We would like to thank the reviewers for their insightful comments and suggestions. We list our detailed point-to-point responses as below. The comments are shown in italic and our responses are shown in blue. Changes in the manuscripts are shown in blue. Figures in this response are numbered as Figure R1, Figure R2, etc.

Reviewer 1:

Overall: The authors describe a method for detecting fusion transcripts in single cell transcriptome data, which they implemented in a tool called 'scFusion'. This method involves first leveraging STAR to align scRNA-seq reads to the genome to identify chimeric reads, and then after identifying candidate fusion transcripts, advanced computational methods are leveraged for filtering likely artifacts. At the heart of scFusion are two methods for fusion filtering: a statistical method leveraging a zero-inflated negative binomial model to describe the background (null distribution) of fusion supporting reads as a function of fusion gene partner expression levels and GC content, and a separate neural network to filter chimeric read alignments based on breakpoint sequence composition to differentiate likely artifacts from randomly fused sequences. The authors apply scFusion and alternative bulk-rna focused methods to single cell transcriptomes in various contexts: simulated scRNA-seq, spiked-in scRNA-seq, and publicly available cancer-relevant single cell rna-seq data sets, and the authors argue that scFusion outperforms alternative methods in each scenario.

Fusion detection in single cell transcriptomes is an important and challenging problem, and the algorithmic developments underlying scFusion are well-intentioned, but I suspect that several experimental design considerations were misguided and deserve further consideration.

Response: We sincerely thank you for your careful review and helpful comments that help us to improve the paper. We designed new experiments to evaluate the algorithms. Please see below for details of the new experiments.

Major critiques:

- 1. The competing bulk rna-seq fusion detection methods were evaluated based on their default parameterizations, which would in most cases not be well suited to single cell transcriptomes. Most methods require at least two to three reads supporting a fusion for a prediction. For peak sensitivity with single cell data, it would be necessary to run the bulk methods with parameters tuned for improved sensitivity. It would appear that scFusion requires only one fusion read to support a prediction for an individual cell (based on the supplementary data provided), which may largely explain its improved sensitivity in certain contexts (e.g., improved sensitivity for IgH-WHSC1 detection).*

Response: Thanks for the suggestion. We did not tune the parameters of bulk methods because we believe that default parameters are the most commonly used parameters for most practitioners. Following the reviewer's suggestion, we try to tune the parameters of bulk methods to improve their sensitivity. FusionCatcher does not have tuning parameters and so we still run FusionCatcher with its default setting. EricScript has only one tuning parameter, the minimum number of supporting reads, and we tune this parameter to improve its performance. Arriba and STAR-Fusion have many tuning parameters including the minimum number of supporting reads. Since it is difficult to find the best combinations of the tuning parameters, we also only tune the minimum-supporting-reads parameter for Arriba and STAR-Fusion. For EricScript, we only tune the minimum supporting reads parameter

from 2, because it is computationally very demanding to run EricScript when setting this parameter as 1.

We first tune the parameter in the newly designed simulation (see our response to comment 3). As expected by the reviewer, the bulk methods have improved sensitivities with a smaller minimum-supporting-reads parameter (Figure R1a). However, their precisions tend to decrease when the minimum-supporting-reads parameter is smaller (Figure R1b). In terms of the F-score, the bulk methods perform the best when the minimum-supporting-reads parameter is 3 (Figure R1c). Similarly, in the real data, when the minimum-supporting-reads parameter is set smaller, bulk methods tend to detect more *TRAJ33/TRAJ12-TRAV1-2* or *IgH-WHSC1* fusions (Figure R2a, b), indicating their higher sensitivities. However, they also tend to detect much more fusions and hence potentially have more false positives (Figure R2c, d). For example, in the T cell data, compared with its default parameters, STAR-Fusion is able to detect more *TRAJ33/TRAJ12-TRAV1-2* recombinations (increased to 71 from 62) but reports much more fusions (from 87 to 354) when the minimum-supporting-reads parameter is set as 1. For the MM data, even we set the minimum-supporting-reads parameter as 1, STAR-Fusion, EricScript and Arriba report the same numbers of *IgH-WHSC1* fusions as with their default parameters. According to this analysis, we now set the minimum-supporting-reads parameter as 3 for EricScript, STAR-Fusion and Arriba.

Figure R1: The recalls, precisions, and F-scores of three bulk methods when tuning the minimum-supporting-reads parameter. This analysis is performed for the simulation setup with 1,000 cells and 4 million reads per cell.

Figure R2: (a, b) The numbers of cells with *TRAJ33/TRAJ12-TRAV1-2* (a) and *IgH-WHSC1* fusions (b) when tuning the minimum-supporting-reads parameter. The dashed lines are the cell numbers reported by scFusion with the fusions, respectively. (c, d) The numbers of fusions reported in T cell data (c) and MM data (d) when tuning the minimum-supporting-reads parameter. The dashed lines are the total number of fusions reported by scFusion. Note that we tuned the parameter for EricScript from 2. So, no data are available for EricScript when the parameter is 1.

2. *The way the authors score false positives and compute specificity is troubling. For simulated data, it is straightforward to identify false positives because the authors have control over which fusions are simulated. However, in the simulations performed by the author, the original chimeric reads from single cell experiments were added to the simulated data as a way of including artifact reads. The problem here is that there are very likely to be real fusions supported by those chimeric reads in addition to true artifacts, and fusions called based on those chimeric reads may in some cases be true fusions and not false positives. Those fusions being called that appear to be neighboring genes and potentially cis-spliced, or involving local rearrangements, are all least likely to derive from artifacts and could be reconsidered.*

Response: We agree with the reviewer that some true fusions could be introduced in the simulated data. We designed a new simulation to avoid this potential problem. Please see our response to comment 3 for details of the new simulation. On the other hand, in the original simulation, we believe that the spike-in chimeric reads obtained from immune cells might contain some true fusions, but the number should be small, because the immune cells are normal cells and thus should not have many true fusions. Furthermore, before adding these chimeric reads, we excluded the chimeric reads from IGH related genes, V(D)J recombinations and neighboring genes.

3. *Rigorous benchmarking was performed in only certain limited contexts. From the simulations (Figure 3), it is clear that STAR-Fusion and Arriba have high sensitivity that surpasses that of scFusion, but both are penalized for having low precision, and I strongly suspect this is because of the addition of the chimeric read supplements from the real single cell data and certain of these putative false positives are likely to be true fusions present in the genuine single cell data. The authors should include experiments performed with simulated data that are not supplemented with real data to avoid this issue.*

Response: Following the reviewer’s suggestion, we perform a new simulation. Similar to the original simulation, we first use FusionSimulatorToolkit to generate RNA-seq data with 150 simulated fusions at various gene expression levels. Then, we design a new method to add technical chimeric artefacts to the simulated data. Briefly speaking, we randomly sample transcript pairs to generate technical fusions between the sampled transcript pairs. The sampling probabilities of the transcript pairs depend on their sequence features to mimic the random annealing and mis-priming in PCR amplification¹. After a technical fusion is introduced to a cell, we further assign a random expression to the technical fusion and add short reads from the fusion to the cell.

More specifically, given any two transcripts T_1 and T_2 (or their reverse complements) and their potential breakpoints b_1, b_2 , we define a mis-priming likelihood

$$A(T_1, b_1, T_2, b_2) = \frac{1}{1 + \exp(180 - 3\psi(T_1, b_1, T_2, b_2))}$$

where $\psi(T_1, b_1, T_2, b_2)$ is the “binding energy” between T_1 and T_2 near the breakpoints b_1, b_2 . Here, we define

$$\psi(T_1, b_1, T_2, b_2) = \sum_{i=1}^6 \phi(S_{1i}, S_{2i}),$$

where $S_1 = S_{11} \cdots S_{16}$ ($S_2 = S_{21} \cdots S_{26}$) is the 6-mer subsequence of $T_1(T_2)$ near $b_1(b_2)$, $\phi(A, T) = \phi(T, A) = 12$, $\phi(G, C) = \phi(C, G) = 21$ and $\phi(N_1, N_2) = 0$ for all other nucleotide pairs (N_1, N_2) . Then, using $A(T_1, b_1, T_2, b_2)$ as the weighting probability, we randomly generate a library of 2.2 million potential technical chimeric sequences. If (T_1, b_1, T_2, b_2) is sampled, the corresponding chimeric sequence is chosen as $T_{11}T_{22}$, where T_{11} is the subsequence of T_1 upstream of b_1 and T_{22} is the complementary of the subsequence of T_2 downstream of b_2 . Then, for each cell, we randomly select chimeric sequences from this library with a new sampling weight

$$W(T_1, b_1, T_2, b_2) = \frac{\psi(T_1, b_1, T_2, b_2)}{1 + \exp(195 - 3\psi(T_1, b_1, T_2, b_2))}$$

and generate chimeric short reads from the selected sequences. The expressions of the technical chimeric sequences are randomly sampled from 1 to 100. The total number of chimeric reads for each cell is set as around 1% of the total number of reads, at the similar level as real scRNA-seq data.

Figure R3 (Figure 3 in the current manuscript) shows the precisions and recalls of scFusion and the bulk methods. We clearly see that scFusion has higher precisions and F-scores than the bulk methods while achieving similar recalls. For example, in the simulation scenario of 1,000 single cells and 4 million reads per cell, the precision and F-score of scFusion are 0.91 and 0.93, respectively. In comparison, the precisions of the bulk methods are only 0.27-0.48 and the F-scores are 0.41-0.64. STAR-Fusion is the best performing bulk method, especially in terms of its precision. This is consistent with real scRNA-seq data analysis, where STAR-Fusion often detects fewer fusions than other bulk methods but is able to detect more fusions such as *TRAJ33-TRAV1-2* and *IgH-WHSC1* that are likely to be true fusions.

Figure R3: The precisions and recalls of scFusion and four bulk methods in six different simulation setups.

4. It is curious that the authors focused their primary findings based on T-cell receptor arrangements in immune cells and IGH fusions in multiple myeloma. There are specialized methods that focus on T-cell receptor configurations (such as TraCeR or TRUST) and one doesn't typically use a chimeric

gene finder for this purpose. In regard to IGH-fusions, these tend to be especially challenging to identify in comparison to most fusion transcripts relevant to cancer (ie. most PCAWG fusions). The authors targeted six publicly available single cell RNA-seq data sets, yet only focus the neural network chimeric read filtering accuracy on all six. Little attention is given to other fusions more typical of fusion-finding efforts in transcriptomes. While one of the single cell data sets corresponds to prostate cancer, no discussion was provided in the paper regarding how well the hallmark *TMPRSS2-ERG* fusion was identified by scFusion and alternative methods. A more comprehensive look at fusion-finding capabilities of scFusion in comparison to alternatives is needed.

Response: The prostate dataset mentioned by the reviewer is the scRNA-seq data of the cancer cell line LNCaP including 288 cells. It is known that LNCaP does not have the *TMPRSS2-ERG* gene fusion². We apply scFusion and the bulk methods to detect gene fusions in this data and compare the detected fusions with the fusions of the LNCaP listed in the Cancer Cell Line Encyclopedia (CCLE) database (<https://sites.broadinstitute.org/ccle/datasets>) (Figure R4a or Supplementary Fig. 6a). As expected, no *TMPRSS2-ERG* fusion is reported by any method. scFusion detects 5 fusions and all are listed in the CCLE database. The bulk methods reported much more fusions but only less than 23% of the fusions are listed in the CCLE database, indicating their potential high false discovery rates. Three fusions detected by the bulk methods and listed in the CCLE database are not detected by scFusion because their p-values do not achieve the significance level. To more comprehensively evaluate scFusion, we further consider two more datasets: a recently published prostate dataset³ and a newly sequenced spike-in dataset with more spike-in fusions.

The recently published prostate data³ has 922 epithelium single cells from 14 patients, but the sequencing length is only 38 bp and thus it is more difficult for fusion detection. scFusion detected 27 cells with the *TMPRSS2-ERG* fusion, slightly more than other bulk methods (Figure R4b or Supplementary Fig. 6c). scFusion reported 10 fusions, much fewer than bulk methods (Figure R4c or Supplementary Fig. 6b). The *TMPRSS2-ERG* fusion can cause the overexpression of the *ERG* gene⁴. This is consistent with our observation that the *ERG* gene is significantly upregulated in the 27 cells reported having the fusion by scFusion (Figure R4d and Supplementary Fig. 6e), indicating that the *TMPRSS2-ERG* fusion is a true fusion. Please see page 8 of the manuscript for more details of this analysis.

Figure R4: (a) The numbers of reported fusions (light blue) and fusions listed in the CCLE fusion database (red). (b) The numbers of cells with *ERG-TMPRSS2* fusion. (c) The numbers of fusions detected. (d) The volcano plot of the differential expression genes of cells between cells with and without the *TMPRSS2-ERG* fusion.

We performed a new spike-in experiment with more spike-in fusions. We experimentally transfect 27 fusions to the 293T cell line (see page 14-15 of the manuscript for details of the experiment). All 27 fusions are from the ICGC pan-cancer gene fusion study⁵, including well-known gene fusions such as *EML4-ALK*, *ROS1-GOPC* and *TMPRSS2-ERG* (Supplementary Table 1). In total, we sequence 729 single cells. Most methods are able to detect these 27 fusions (Figure R5a or Figure 5a), but the total number of fusions reported by scFusion is much fewer than other methods (Figure R5b or Figure 5b). In total, 86.5% of the fusions detected by scFusion are in the 27 spike-in gene fusions or are supported by two or more chimeric reads in the bulk sequencing of the 293T cell line, much higher than bulk methods (Figure R5c or Figure 5c).

Figure R5: (a) The numbers of reported spike-in fusions. (b) The numbers of reported fusions. (c) The proportions of fusions having bulk supporting chimeric reads or in the 27 spike-ins.

We agree with the reviewer that V(D)J recombinations are commonly not analyzed by fusion detection tools and the *IgH*-fusions are more challenging than other fusions. We would like to emphasize that, similar to gene fusions, V(D)J recombinations also generate chimeric reads. Theoretically, these recombinations can also be analyzed by fusion detection tools, but probably is more challenging. The good performance of scFusion in detecting these challenging fusions demonstrates that scFusion can accurately detect a wide range of fusions and hence can be used in more applications.

5. For the simulations leveraging the Fusion Simulation Toolkit, from the methods description it isn't clear how the data were simulated for single cells, given that the toolkit was developed toward simulating bulk data. Was each cell targeted and modeled separately via the read simulator? Or was a single model created and run *N* times to generate *N* cells worth of scRNA-seq data? Ideally, it's the former.

Response: Sorry about the confusion. A single model was created and run *N* times to generate *N* cells. To see the difference between the single-model and multiple-model approaches, we simulate scRNA-seq data of 500 cells with 4 million reads per cell using both approaches. We find that the performances of all methods are very similar (Figure R6). Since training an individual model for each single cell is computationally very demanding, we still use the single-model approach and clarify it in the paper.

Figure R6: The precisions and recalls of the fusion detection methods for the two simulation datasets generated by the single-model and multiple-model approaches.

6. *Extended data figure 1 shows the impact of the filtering methods implemented in scFusion and would best be shown as a main figure instead of being restricted to the supplement. From this figure, it is clear that the statistical model is most responsible for the selection of candidate fusions, with only small contributions from the neural network towards filtering potential artifacts. This should be made clear in the main text of the manuscript.*

Response: Following the reviewer’s suggestion, we move this figure to Figure 4b. We also clarify in the paper that the most artefacts are filtered by the statistical model (page 5 of the manuscript).

7. *Finally, it is curious that even after applying the advanced computational methods for filtering candidates, there continues to be additional 'two-many ...' filters (extended data figure 1), and what appears to be 'badgene' listing that assists in additional filtering (found in the code). It isn't clear to me what extent these additional 'badgene' filters contribute to overall fusion prediction capabilities, and this should be made clear in the manuscript as well.*

Response: Thanks for your careful review and sorry for the confusion. We did not use the badgene filter. The badgene folder was created in an earlier version of scFusion and we forgot to delete it when releasing scFusion Version 1.0. Now we removed the badgene folder in the latest version of scFusion. The too-many-partner-filter and the too-many-discordant-filter are added to filter potential false positives that are likely generated due to incorrect alignments of short reads (e.g. short reads from genes with homologous sequences). This was made clear in the manuscript (page 3 of the manuscript).

Comments regarding the software:

8. *I encourage the authors to create a docker image containing all needed software so it can be easily run as a single package, without requiring additional setup time on the part of the user.*

Response: Thanks for the great suggestion. Following the suggestion, we created the docker image for scFusion (see README on GitHub about how to acquire the docker image).

9. *The hg19 mappability file could be included with the software instead of requiring the user to download it separately from the ucsc track downloader.*

Response: Thanks for the suggestion. The hg19 mappability file was added to the data folder in the latest version of scFusion.

Reviewer 2:

Overall: The manuscript entitled “Single cell gene fusion detection by scFusion”, by Dr. Zije Jin and colleagues, describes scFusion, a software tool for detecting gene fusions in full-length single cell gene expression datasets. The tool uses a combination of statistical modeling and machine learning (ML) to identify true fusion events from a high level of data noise.

This work is of potentially high significance, as scFusion appears to be the only method to date designed for fusion detection in single-cell data, although the authors don’t state this clearly. Based on the evaluations included in the study, the method has higher sensitivity and accuracy than other, bulk RNA sequencing based approaches. It also runs faster (although still consuming 100s of hours of CPU time per analysis). Despite these strengths, the manuscript has a number of weaknesses, as detailed below.

Response: We highly appreciate your encouraging comments. We sincerely thank you for your great suggestions that help us to improve the manuscript.

Biological significance:

1. *The authors utilize published datasets, with the exception of a dataset generated for addressing a technical point, therefore the manuscript reports no biologically novel data.*

Response: We agree with the reviewer that this manuscript does not report biologically novel data. This work is mainly about developing a single cell fusion detection algorithm. Hence, we only generated a scRNA-seq data for algorithm evaluation. In the revised manuscript, we generated a larger fusion-spiking-in data to evaluate the algorithm more comprehensively, and again confirmed that scFusion had better performance than bulk methods. Please see our response to your comment 7 for more details of the new data. In the future, by collaborating with cancer biologists, we plan to profile single cell data from cancer patients (such as prostate cancer patients or lung cancer patients) to more systematically analyze gene fusions at the single cell level.

2. *Similarly, the published datasets are used for evaluating the method, without any essential new insight.*

Response: Because this work focuses on developing a fusion detection algorithm for single cells, all datasets are used for evaluating the method. The proposed algorithm is the first single cell fusion detection algorithm. Although the paper does not report much new biological findings, considering the importance of fusions in tumors and scFusion’s high sensitivity, precision, and computational efficiency, we believe that this manuscript provides an essential technical advance for tumor studies.

Presentation:

3. *The presentation is often unclear, to the point where it simple prevents unambiguous understanding of the material. For example, I was simply not able to clearly interpret the statement “Finally, scFusion filters out fusion candidates whose number of supporting discordant reads is ten times more than the supporting split-mapped reads and the candidates with a partner gene involving in more than five fusion candidates.”. The clarity of the presentation must improve to make this manuscript accessible for the authors (not to mention the reviewers).*

Response: We are sorry for the unclear presentation. We have rephrased this sentence to “At last, scFusion applies two more filters to filter potential false positives that are likely generated by incorrect alignments of short reads (e.g. short reads from genes with homologous sequences). If a fusion’s supporting discordant reads are more than 10 times of its supporting split-mapped reads, scFusion filters the fusion candidate, or, if a gene is in more than 5 fusion candidates, we filter all fusion candidates involving this gene.” Note that a paired-end read is a fusion’s supporting discordant read if its two ends are respectively mapped to the two partner genes of the fusion. We also proofread the manuscript and clarify all unclear descriptions.

4. *Along similar lines, the description of the methods is restricted to mathematical details and is therefore inaccessible. I could gain very little insight into the inner workings of the algorithms.*

Response: Sorry for the unclear description. We added more descriptions of the mathematical method in the manuscript to make it more accessible to general readers. We hope that this will make the method description clearer.

In the “Statistical Model” section (page 10-11 of the manuscript), we added the following descriptions before giving the less accessible mathematical descriptions (but probably more rigorous). “Suppose that there are n cells and N fusion candidates. Let y_{ij} be the number of split-mapped reads supporting a fusion candidate i in a cell j . The fusion candidate i can either be a true fusion or from the background noises. If the distribution of the background noises is known, we can perform a statistical test to test if the observed numbers of supporting reads y_{ij} ’s ($j = 1, \dots, n$) fit well to the distribution of the background noise. Candidates with significant p-values would be more likely to be true fusions. However, we do not know the distribution of the background noises and hence we need to first estimate this distribution. By assuming that the distribution of the background noises is a zero-inflated negative binomial (ZINB) distribution, estimating the distribution can be achieved by estimating the parameters in the ZINB distribution. Since we observe that the number of supporting chimeric reads depends on covariates such as the local GC content and expressions of partner genes (Fig. 2a, b), we establish a regression model to link the parameters in the ZINB distribution and the covariates. The parameters of the regression model can be estimated by the maximum likelihood estimation.”

In the “Statistical Test for Significant Fusions” (page 11-12 of the manuscript), we added the following description before the mathematical descriptions. “Plugging-in the MLE estimates of the regression parameters, we can estimate the background noise distribution for each fusion candidate. With the distribution estimate, we can test if the observed number of supporting chimeric reads for the fusion candidate is likely to be sampled from the background distribution. If the background distribution is unlikely to generate an observation that are larger than the observed supporting reads number, or in other words, if we obtain a very small p-value, we reject the null hypothesis that the fusion candidate is from the background noise and retain the fusion as a true fusion candidate.”

Technical issues:

5. *There are some unjustified methods in generating the simulated datasets use in the tool evaluations. For example, the method of generating a positive training dataset by concatenating random pairs of short reads does not appear to faithfully represent true fusions. As another example, the authors mixed “real” fusion transcripts with their simulated background data, which is potentially problematic. Especially when using machine learning, it is entirely possible for the ML algorithm “learns” to identify such spike-ins based on extraneous differences between the simulated and the real sequencing reads, rather than recognizing the fusion. This effect could be potentially quantified by spiking in additional “real” data without fusions.*

Response: Thanks for the great suggestion. Following the reviewer’s suggestion, we evaluated the performance of the machine learning algorithm by spiking-in additional T cell data without fusions. More specifically, we created synthetic chimeric reads from random pairs of short reads from the T cell data that were not chimeric. We then compared the artefact scores of these synthetic chimeric reads with those of chimeric reads in the T cell data. The machine learning algorithm was trained using the original simulation data. 95% of the synthetic chimeric reads have artefact scores less than 0.5, while > 90% of the real chimeric reads have artefact scores greater than 0.9 (Figure R7). It is thus unlikely that the machine learning algorithm learned to identify spike-ins based on extraneous difference between the simulated and the real sequencing reads. Furthermore, to entirely avoid this problem, we performed a new simulation (see below).

Figure R7: The density of artificial scores of chimeric reads and synthetic chimeric reads using the machine learning model trained based on the original simulation data.

Similar to the original simulation, we first use FusionSimulatorToolkit to generate RNA-seq data with 150 simulated fusions at various gene expression levels. Then, we design a new method to add technical chimeric artefacts to the simulated data. Briefly speaking, we randomly sample transcript pairs to generate technical fusions between the sampled transcript pairs. The sampling probabilities of the transcript pairs depend on their sequence features to mimic the random annealing and mis-priming in PCR amplification¹. After a technical fusion is introduced to a cell, we further assign a random expression to the technical fusion and add short reads from the fusion to the cell. Details of the new simulation is in page 13-14 of the revised manuscript.

Figure R8 (Figure 3 in the current manuscript) shows the precisions and recalls of scFusion and the bulk methods in the new simulation. We clearly see that scFusion has higher precisions and F-scores than the bulk methods while achieving similar recalls. For example, in the simulation scenario of 1000 single cells and 4 million reads per cell, the precision and F-score of scFusion are 0.91 and 0.93, respectively. In comparison, the precisions of the bulk methods are only 0.27-0.48 and the F-scores are 0.41-0.64. STAR-Fusion is the best performing bulk method, especially in terms of its precision. This is consistent with the real scRNA-seq data analysis, where STAR-Fusion often detects fewer fusion candidates than other bulk methods but is able to detect more important fusions such as *TRAJ33-TRAV1-2*, *TRAJ12-TRAV1-2* and *IgH-WHSC1*.

Figure R8: The precisions and recalls of scFusion and four bulk methods in six different simulation setups.

6. *The argument that scFusion works better than other methods in the T-cell dataset simply because it calls less fusions than the other methods and arguing that these extra detected fusions must be false positives is circular.*

Response: In the T-cell dataset, scFusion found fewer fusions than bulk methods, but was also able to detect V(D)J combinations (*TRAJ33-TRAV1-2*, *TRAJ12-TRAV1-2*) that most bulk methods failed to detect. In fact, except STAR-Fusion, all other bulk methods did not detect any V(D)J recombinations, even though they found 10-500 times more fusions than scFusion (Figure R9a or Figure 6a). For STAR-Fusion, it only identified one of the two V(D)J recombinations that scFusion reported (*TRAJ33-TRAV1-2*), and it reported a smaller number of cells with this recombination than scFusion (Figure R9b, c or Figure 6e, f). So, we think although scFusion reported much fewer fusion candidates, it was able to detect more true fusions than bulk methods.

Figure R9: (a) The number of detected gene fusions by the five methods. (b, c) The barplots of numbers of cells with the *TRAJ33-TRAV1-2* and *TRAJ12-TRAV1-2* recombinations by different algorithms.

7. *Related to the previous point, if all detected fusion candidates (in addition to the spike-ins) are indeed false positives, the scFusion false discovery rate (FDR) would be $10/(10+4)$, over 70%*

considering 10 false positives and 4 true positives. This is potentially too high for practical application.

Response: The 293T cell line that we used for the spike-in experiment might contain gene fusions. This is why we also performed a bulk RNA sequencing of the cell line. Viewing all fusion discoveries other than the five spike-in fusions would very likely overestimate the false discovery rate. If we also view fusion candidates with supporting chimeric reads in bulk data as true positives, the FDR of scFusion is about 21%. Furthermore, we performed a new spike-in experiment with more spike-in fusions (27). scFusion totally detects 37 fusions including all the spike-in fusions. In this dataset, if we view the non-spike-in fusions as false positives, the FDR of scFusion is 27%. If we also view fusion candidates with supporting bulk chimeric reads as true positives, the FDR of scFusion is about 13.5% (Figure R10). Please see page 5 and 14 of the main manuscript for more details of this new spike-in data.

Figure R10: The FDR of each method. FDR is defined as the proportion of fusions that are not in the 27 spike-in fusions or not having supporting bulk chimeric reads.

Reviewer 3:

Overall: Within the manuscript “Single cell gene fusion detection by scFusion”, the authors detail a new single cell sequencing computational tool that precisely detects gene fusions from RNA-seq data. They describe the development of scFusion, in addition to demonstrating the utility of the tool using a combination of simulated, spiked and publicly available cancer datasets.

In contrast to other approaches – which repurpose computational tools for bulk RNA-seq analysis – the development of scFusion represents the first analytical tool specifically designed to detect gene fusions from a single cell source. As such, scFusion is a welcome addition to the current methods available for detection and analysis of gene fusions in cancer patient samples.

However, there are a few key issues relating to the manuscript that require elaboration or amendment.

Response: We appreciate the reviewer’s encouraging comments and valuable suggestions that help us improve the manuscript. The questions are addressed below.

Methods comment:

1. “Gene fusion spike-in and single cell sequencing.” Why did the authors approach the experiment in this way? Attempting to transfect cells with multiple constructs at the same time is a very

random approach. Why did the authors not transfect separate 293T cell cultures independently with a single construct, pool successfully transfected cultures at known proportions and then use this as input for scRNA-seq? This would provide more accurate knowledge of the starting material and provide a blueprint upon which to design further assays to change the titration amounts of each transfected cell and precisely determine the lower detection limits of scFusion. With the current approach, the authors can only state in the Results section that “scFusion did not detect the spiked-in fusion CCDC6-RET, since the fusion occurred in only two cells and did not achieve the significance level.”, which gives no indication of the true detection limit of the computational tool.

Response: Thanks for the great suggestion. Following the reviewer’s suggestion, we create a new spike-in data with 27 fusions to evaluate the algorithm more comprehensively. The spike-in genes were independently transfected into separate 293T cell cultures. Then, we used fluorescence-activated cell sorting (FACS) to select 27 cells from each culture of the 27 cell cultures and performed scRNA-seq. The spike-in experiment is detailed below.

cdNA sequences of 27 fusion genes were synthesized and constructed into independent lentiviral vectors. Every lentiviral vector along with two auxiliary packaging plasmids was co-transfected into independent 293T cells. After 48 hours, the supernatant was collected from the 293T cells and filtered through a 0.45 μ m membrane. The 27 recombinant lentiviral particles containing the target fusion genes and the green fluorescent protein (GFP) reporter gene were collected. Then, the 27 recombinant lentivirus particles were infected into 293T cells, respectively. 72 hours after infection, the medium was changed and the expression of GFP in cells was checked under a fluorescence microscope to determine if lentivirus infection was successful. After the infection, 27 cell cultures expressing different target fusion genes were collected. The collected cells were washed with 1x PBS and resuspended with 2mL PBS. Single cell sorting was performed on the BD Biosciences FACS-ARIA platform and the single GFP positive cells (27 cells from each of the 27 cell cultures) were screened into 96-well PCR plates, respectively, for the next step of single cell RNA library construction. Poly(A)-transcripts of total RNA of single cells were reverse transcribed and amplified using the SMART-seq2 protocol. The amplified cDNA was tagged by Nextera XT kit (Illumina) and libraries were sequenced by NovaSeq (Illumina). The 293T cells were purchased from the National Infrastructure of Cell Line Resource (<http://www.cellresource.cn/>).

In total, we obtained scRNA-seq data of 729 single cells. scFusion identified all the 27 spike-in gene fusions and 10 other fusions (Figure R11a or Figure 5a). Most bulk methods also detected all 27 spike-in fusions, but they reported much more fusions (Table S2-6, Figure R11b, or Figure 5b), indicating their potentially high FDRs. We also performed bulk RNA-seq of the 293T cell line. 86.5% (32) of fusions detected by scFusion had 2 or more supporting chimeric reads in the bulk data or are the spike-in fusions, much higher than bulk methods (2.7%-21.4%, Figure R11c or Figure 5c), indicating that the FDR of scFusion is much lower.

Figure R11: (a) The numbers of spike-in fusions reported by five methods. (b) The numbers of all fusions reported by five methods. (c) The proportions of fusions having bulk supporting chimeric reads.

Results comments:

1. For the scFusion method, the authors state that, “Finally, scFusion filters out fusion candidates whose number of supporting discordant reads is ten times more than the supporting split-mapped reads and the candidates with a partner gene involving in more than five fusion candidates.”. Could the authors detail how they defined / assigned these seemingly arbitrary numerical values?

Response: For a pair of paired-end reads with an insert size 300bp and a read length 100bp sampled from around the breakpoint of a fusion, the probability that the breakpoint locates at the two reads is roughly two times of the probability that the breakpoint locates between the two reads (because the total length of the two reads is 200bp and the unsequenced part is 100bp). Thus, for a gene fusion, its number of supporting discordant reads should be largely 1/2 of its supporting chimeric reads. Therefore, we set a rather stringent cutoff 10 for the ratio between supporting discordant reads and supporting chimeric reads. We agree with the reviewer that this cutoff is a bit arbitrary, but this cutoff does not have much influence on the final fusion list. For example, if we determine the cutoff at the 98% quantile of the discordant-vs-split-mapped supporting reads ratios of all fusion candidates, the number of detected fusions are the same for all data analyzed except the multiple myeloma data and prostate patient data (Figure R12). Note that we add two more scRNA-seq data in the revised manuscript (the LNCaP and the prostate patient data). Details of the two additional datasets are in page 7-8 of the manuscript. The extra fusions in MM data are all related to *IgH* genes. In the prostate patient data, three more fusions are filtered using the 98% quantile cutoff since the 98% quantile of the discordant-vs-split-mapped supporting reads ratios is zero and all fusions with any discordant supporting reads are filtered. We therefore still use 10 as the default cutoff and provide users an option to select the cutoff based on the quantile. The too-many-partner filter was used by bulk methods such as STAR-Fusion and we followed their setup.

Figure R12: The numbers of reported fusions by scFusion applying different discordant-vs-split-mapped supporting reads ratio.

2. *For the simulated data, the authors state that, “The fusions missed by scFusion are mostly fusions with low expression levels (Extended Data Fig. 5).”. Could the authors please comment on whether this could be an additional limitation for scFusion if it was applied to scRNA-seq data from a highly heterogeneous tumour tissue sample?*

Response: scFusion has a reduced sensitivity for fusions with low expression. This does pose a limitation for detecting fusions in highly heterogeneous tumor samples. If a fusion only presents in a small subclone and the fusion is lowly expressed, scFusion will have limited power detecting the fusion. On the other hand, if a fusion presents in a large proportion of tumor cells, even if the fusion is lowly expressed, because of its large number of supporting cells, scFusion would still have enough power to detect the fusion. We added more discussions about this limitation in the discussion section of the manuscript (page 9 of the manuscript).

Discussion comment:

1. *The authors conclude with, “Analysis of gene fusions at single cell level will provide unprecedented opportunities to study their roles in tumor development, tumor heterogeneity as well as tumor cell’s responses to various pharmaceutical therapies.”. However, the computational tool has only been applied in vivo to examples of multiple myeloma, pancreatic cancer (which does not harbour disease-associated gene fusions) and artificially spiked cell line datasets. Without further demonstrated application of scFusion to other blood cancer and solid tumour datasets, I find this statement somewhat overreaching.*

Response: We deleted this sentence in the new manuscript.

General comment:

1. *There are small spelling and grammatical errors throughout the submission that should be resolved.*

Response: We carefully checked the manuscript and corrected typos and grammatical errors.

Reference

- 1 Haas, B. J. *et al.* Chimeric 16S rRNA sequence formation and detection in Sanger and 454-pyrosequenced PCR amplicons. *Genome research* **21**, 494-504, (2011).
- 2 Mani, R.-S. *et al.* TMPRSS2–ERG-mediated feed-forward regulation of wild-type ERG in human prostate cancers. *Cancer research* **71**, 5387-5392, (2011).
- 3 He, M. X. *et al.* Transcriptional mediators of treatment resistance in lethal prostate cancer. *Nature Medicine* **27**, 426-433, (2021).
- 4 Adamo, P. & Ladomery, M. R. The oncogene ERG: a key factor in prostate cancer. *Oncogene* **35**, 403-414, (2016).
- 5 Campbell, P. J. *et al.* Pan-cancer analysis of whole genomes. *Nature* **578**, 82-93, (2020).

Reviewers' Comments:

Reviewer #1:

Remarks to the Author:

Thanks for the opportunity to re-review the manuscript "Single cell gene fusion detection by scFusion" by Jin et al. I appreciate the responses to the reviewers' comments from the first review. While the manuscript has been improved as a result of the review process, I continue to find issues that the authors should address, as detailed below.

Most of my critiques relate to the benchmarking methods used by the authors.

For the initial experiment performed using simulated data, it isn't clear where the low levels of precision are coming from for the various competing methods - ie. what the false positives correspond to. The authors didn't include the prediction results as supplementary data, and the simulated scRNA-seq data were not made available - both problems that should be resolved. With simulated data, it's usually difficult to find high false positives, and those false positives that do exist tend to correspond to alignment mis-mapping events. Since the authors are requiring all methods to predict fusions in at least 2 cells to be scored as true or false positives, it isn't clear how randomly generated chimeric background noise could contribute to false positive predictions, as I would think that it would be highly unlikely for the same chimeric pair to contribute towards predictions in multiple cells and hence contribute towards false positives. It would also be useful to better understand what specific algorithmic component of scFusion is contributing to excluding such false positives in these cases as well. Since scFusion uses STAR chimeric alignments to identify initial candidate fusions - as does STAR-Fusion and Arriba - there must be some specific aspect of scFusion's statistical model or bi-LSTM that is filtering out these predictions as unfit. With simulated data, it should be possible to specifically define those characteristics responsible - showing that several of the STAR-Fusion or Arriba falsely predicted fusions are also initially defined as candidates by scFusion, and subsequently filtered due to the statistical model or neural network predictions, and this would add important insights and increase transparency in the manuscript.

While exploring the prediction results for the second experiment described involving spiked-in fusions, I noticed several issues that should be addressed. First, instead of calling individual fusion gene pairs as false positives, each and every predicted isoform of a given fusion gene pair was being called as false positives. For example, in TableS3, Arriba is shown to predict the fusion LRRFIP2--CAV3 with 7 different breakpoints, and so instead of calling one false positive for that fusion, it's counted 7 times, which seems extreme. Approximately 20% of the false positives for Arriba are attributed to this feature. Hundreds of additional false positives involve gene pairs involving the same gene (self-fused), and so would represent an intra-gene effect and not a fusion pair. Over a hundred involve neighboring genes that presumably involve cis-splicing events. The false positives include the gene pair KANSL1--ARL17 predicted by Arriba, STAR-Fusion, and FusionCatcher and a well known 'normal' fusion found in about 30% of those with European ancestry due to a natural structural rearrangement (see PMID:28881586). It isn't clear why scFusion is not reporting many these fusions, nor why such fusions would be scored as false positives. All of these issues contribute to scFusion appearing more accurate and the comparators less accurate and are ill justified.

The authors indicate "The reference genome was always chosen as hg19. For the bulk methods, we also applied the same ad hoc filters of scFusion to make different algorithms comparable. Specifically, we applied the pseudogene-, lncRNA-, noapproved-symbol-gene-, intron-, too-many-partner- and too-many-discordant- filters to the fusion candidates of bulk methods." However, many LINC- predictions are apparent in the supplementary tables, indicating that identical filtering rules were not applied as described, and the comparators many false positives were being declared for lncRNAs. I haven't carefully examined whether other filtering criteria were being applied.

The fusion filtering criteria applied by scFusion involving removing lncRNAs and those genes without approved symbols²¹ (such as RP11-475J5.6) do not appear to be justified. If there are true cancer-relevant fusions involving such genes, and there are several well known (PVT1 of PVT1--MYC) or recently discovered, they would be excluded without any scientific basis for doing so.

While it is true that scFusion is the first fusion method that is specifically tailored towards predicting fusions in single cell data, the statistical model component of scFusion is the only method that takes into consideration features of single cell RNA-seq data. This should be made explicitly clear in the manuscript.

Minor issues:

Figure 1 (a,b) and Figure 1 (c,d,e,f) are disjoint and should be presented separately to avoid conflating aspects of the statistical model and those of the neural network.

While the neural network appears to have good accuracy at discriminating chimeric alignments from others, it also appears to contribute very little to overall fusion prediction accuracy. It makes me wonder if the chimeric alignments are enriched for certain chimeric breakpoints rather than being more generally informative of chimeric alignments. If the chimeric alignments are not first normalized for chimeric breakpoints (such that chimeric breakpoints are more uniformly represented among the large list of chimeric alignments), perhaps this should be performed so that certain breakpoints aren't greatly biasing the predictions, and the neural network may be able to contribute more to discriminating among fusion predictions in scFusion.

The simulated RNA-seq data used for the initial simulation experiment should be made available in addition to the predictions according to method.

The fusion predictions for scFusion in table S2 sometimes have flipped orientation with respect to the spiked in fusions. ie. NUP214--SET and ROS1--GOPC.

For Figure 5C, it isn't clear how the bulk fusions were defined - please include their definition, results in supplementary data, and be sure the rna-seq data are publicly available if not already.

From my experience, Ericscript has provided little value as a fusion predictor given its high false positive rates. It isn't clear why the authors include Ericscript here as opposed to other tools available.

Thanks for making a docker image available. I highly recommend adding your Dockerfile to github - this provides an unambiguous installation protocol for others.

I did try running your test data through using your docker image and I was not successful:

""""

```
root@e31afd9a60c3:/data# python /usr/local/src/scFusion-1.1/scFusion.py -f `pwd` /testdata -o
`pwd` /testdata/testout -b 1 -e 10 -s /path/to/ref_genome.fa.star.idx -t 8 -g
/ctat_genome_lib/ref_genome.fa -a /path/to/ref_annot.gtf -n 0.9
```

Preparing for scFusion!

Parameter Check Complete!

10 cell sequencing files found!

Start mapping! Index: 1 ~ 10, using core: 7

```
/usr/local/src/scFusion-1.1/bin/StarMapping_Chimeric.sh: 8: /usr/local/src/scFusion-
1.1/bin/StarMapping_Chimeric.sh: Syntax error: Bad for loop variable
Finish mapping! Index: 1 ~ 10
```

Start Basic Processing! Index: 1 ~ 2, using core: 1

Start Basic Processing! Index: 3 ~ 4, using core: 1

Start Basic Processing! Index: 5 ~ 6, using core: 1

Start Basic Processing! Index: 7 ~ 8, using core: 1

Start Basic Processing! Index: 9 ~ 10, using core: 1

```
/usr/local/src/scFusion-1.1/bin/CombinePipeline_before_FS.sh: 15: /usr/local/src/scFusion-1.1/bin/CombinePipeline_before_FS.sh: Syntax error: Bad for loop variable
Finish Basic Processing! Index: 1 ~ 2
```

```
/usr/local/src/scFusion-1.1/bin/CombinePipeline_before_FS.sh: 15: /usr/local/src/scFusion-1.1/bin/CombinePipeline_before_FS.sh: Syntax error: Bad for loop variable
Finish Basic Processing! Index: 3 ~ 4
```

```
/usr/local/src/scFusion-1.1/bin/CombinePipeline_before_FS.sh: 15: /usr/local/src/scFusion-1.1/bin/CombinePipeline_before_FS.sh: Syntax error: Bad for loop variable
Finish Basic Processing! Index: 5 ~ 6
```

```
/usr/local/src/scFusion-1.1/bin/CombinePipeline_before_FS.sh: 15: /usr/local/src/scFusion-1.1/bin/CombinePipeline_before_FS.sh: Syntax error: Bad for loop variable
Finish Basic Processing! Index: 7 ~ 8
```

```
/usr/local/src/scFusion-1.1/bin/CombinePipeline_before_FS.sh: 15: /usr/local/src/scFusion-1.1/bin/CombinePipeline_before_FS.sh: Syntax error: Bad for loop variable
Finish Basic Processing! Index: 9 ~ 10
```

Start Combining!

Finish Combining!

Start Retraining the Neural Network!

Traceback (most recent call last):

File "/usr/local/src/scFusion-1.1/scFusion.py", line 273, in <module>

aaa = subprocess.check_output(

File "/usr/local/anaconda3/lib/python3.8/subprocess.py", line 415, in check_output

return run(*popenargs, stdout=PIPE, timeout=timeout, check=True,

File "/usr/local/anaconda3/lib/python3.8/subprocess.py", line 516, in run

raise CalledProcessError(retcode, process.args,

subprocess.CalledProcessError: Command 'sh /usr/local/src/scFusion-

1.1/bin/CombinePipeline_Retrain.sh /data/testdata/testout . /usr/local/src/scFusion-

1.1/bin/./data/weight-V9-2.hdf5 10 /usr/local/src/scFusion-1.1/bin/' returned non-zero exit status

1.

""

Reviewer #2:

Remarks to the Author:

The resubmission from Dr. Jin and colleagues has been revised in response to reviewer comments. Although as stated in my original review, the work is potentially significant, I did not find that the current resubmission, or the responses to my critiques, were adequately addressed. The remaining major issues are as follows:

(1) Biological significance. As the authors readily admit, the manuscript is purely methodological, the one novel spike-in dataset addressing technical questions. Without a demonstration of how the proposed method detecting fusion transcripts in single-cell RNA sequencing data can drive biological discovery, this study is perhaps better suited for a more technically oriented specialist

journal.

(2) Method descriptions. Despite claims that the methods have been made more accessible, they are still highly mathematical, without giving an insight why the specific distributions and assumptions are appropriate. For example, the authors write "By assuming that the distribution of the background noises is a zero-inflated negative binomial (ZINB) distribution...". But why is the noise best approximated as a zero-inflated negative binomial?

(3) Fusion simulation. In response to my criticism that the fusions that are likely to be biologically present in a sample may not be well represented by randomly putting together reads, the authors address a different problem, i.e. whether the method trained for identifying and rejecting artifactual fusions based on simulated reads also work for identifying and rejecting randomly spliced together "read reads". The original problem still remains unaddressed.

(4) False positive rate assertions. When addressing the issue of specificity (namely that, just by calling a smaller number of fusions, scFusion has a lower false positive rate compared to other methods), the authors start talking about sensitivity. Again, this does not address the original question.

On the positive side, the overall wording of the manuscript is now substantially improved.

Overall, the current revision does not address the critical comments in my original review. Most issues have been sidestepped, or "explained away", rather than addressed, as detailed above.

Reviewer #3:

Remarks to the Author:

The revised manuscript looks much improved, and the authors have addressed all my comments. Some of the grammar used is still inaccurate, and I would advise one final read-through to refine the submission.

Responses to Reviews' Comments

We would like to thank the reviewers for their detailed comments and critiques. We list our responses below. The comments are shown in italic and our responses are in blue.

Reviewer 1:

Thanks for the opportunity to re-review the manuscript "Single cell gene fusion detection by scFusion" by Jin et al. I appreciate the responses to the reviewers' comments from the first review. While the manuscript has been improved because of the review process, I continue to find issues that the authors should address.

Response: We sincerely thank you for the helpful comments that help us improve scFusion and the manuscript. Please see below for the details.

Most of my critiques relate to the benchmarking methods used by the authors.

1. For the initial experiment performed using simulated data, it isn't clear where the low levels of precision are coming from for the various competing methods - ie. what the false positives correspond to. The authors didn't include the prediction results as supplementary data, and the simulated scRNA-seq data were not made available - both problems that should be resolved. With simulated data, it's usually difficult to find high false positives, and those false positives that do exist tend to correspond to alignment mis-mapping events. Since the authors are requiring all methods to predict fusions in at least 2 cells to be scored as true or false positives, it isn't clear how randomly generated chimeric background noise could contribute to false positive predictions, as I would think that it would be highly unlikely for the same chimeric pair to contribute towards predictions in multiple cells and hence contribute towards false positives. It would also be useful to better understand what specific algorithmic component of scFusion is contributing to excluding such false positives in these cases as well. Since scFusion uses STAR chimeric alignments to identify initial candidate fusions - as does STAR-Fusion and Arriba - there must be some specific aspect of scFusion's statistical model or bi-LSTM that is filtering out these predictions as unfit. With simulated data, it should be possible to specifically define those characteristics responsible - showing that several of the STAR-Fusion or Arriba falsely predicted fusions are also initially defined as candidates by scFusion, and subsequently filtered due to the statistical model or neural network predictions, and this would add important insights and increase transparency in the manuscript.

Response: We included the prediction results from one simulation as the supplementary data (Supplementary Table 2-6). The simulated scRNA-seq data was too large (17 terabytes).

We made a small part of the simulation data publicly available at

<https://drive.google.com/drive/folders/1H5esW31rrq7in4jxBwtGMhJ9v5C81-to?usp=sharing>

and provide the source code for generating the simulation data in GitHub

<https://github.com/ZijieJin/SimulationscFusion>.

Previous researches showed that random annealing or mis-priming occurred during PCR amplification¹ could lead to the formation of chimeric artefacts. The mis-priming events are sequence-dependent and sequences with certain features (e.g. sequences with greater sequence homology) will be more likely to generate chimeric artefacts. Current RNA-seq simulation tools cannot simulate these artefacts. Thus, as pointed out by the reviewer, the false positives in

simulation generated by available tools tend to correspond to mis-mapping events. Simulations generated by these tools could under-estimate false positives. To simulate these mis-priming events, we designed a novel method to generate chimeric artefacts. Largely speaking, this method samples pairs of transcripts to generate chimeras and the sampling probabilities depend on the sequences of transcripts. The same chimeric sequences could be introduced to different cells because some pairs of transcripts have much higher probabilities of generating chimeras than other transcript pairs. Please see page 14 in the manuscript or our response to comment 3 in the first round of the review for details of this method.

Following the reviewer’s suggestion, we turned off the statistical model or the deep learning model and evaluated the performance of scFusion without one of these two models (Figure R1). Compared with the statistical model, the deep learning model was less effective in terms of removing false positives, but it still was able to filter a large portion of false positives with minimal influences on the sensitivity. For example, in the scenario with 500 cells and 4 million reads per cell, turning off the deep learning model made the precision of scFusion decreased to 0.75 from 0.918 (Figure R1b), but the sensitivity only increased to 0.9 from 0.87 (Figure R1a). Moreover, if not using the deep learning model, the false discovery rates (FDR) of scFusion could be as large as ~25%, but after applying the deep learning model filter, the FDRs reduced to < 10% (Figure R1c).

To further demonstrate the effect of two models, we extracted the false positives of STAR-Fusion and Arriba and investigated their p-values and the artefact scores given by scFusion (Figure R2). Among these false positives, around 54% and 70% can be filtered by the statistical model (setting p-value > 0.01) and the deep learning model (setting artefact score > 0.75), respectively. The deep learning model could filter many false positives that were not filtered by the statistical model (Figure R2). For example, the fusion candidate *ATG4B-SUMF2* was reported by both STAR-Fusion and Arriba. The fusion had a p-value 10^{-41} and thus could not be filtered by the statistical model, but it had an artefact score 0.953 and was filtered by the deep learning model.

Figure R1: The performance of scFusion when turning off the statistical model or deep-learning model. For comparison, the performance of scFusion is also shown. (a) The sensitivities. (b) The precisions. (c) The FDRs.

Figure R2: The scatterplot of the artefact scores against p-values for the false positives reported by (a) STAR-Fusion and (b) Arriba. The numbers in the figure are the numbers of points in the four different regions. If we set the p-value cutoff as 0.01, 25 and 24 of the false positives predicted by STAR-Fusion and Arriba had significant p-values (<0.01) and thus cannot be filtered by the p-values, but can be filtered by the deep learning model due to their large artefact scores (>0.75). Note that this analysis focuses on false positives reported by the bulk methods. Many false positives have been filtered by the build-in filters in the bulk methods or by the ≥ 2 cell filter. Hence, the deep learning model appears to be more effective than in Figure R1.

2. While exploring the prediction results for the second experiment described involving spiked-in fusions, I noticed several issues that should be addressed. First, instead of calling individual fusion gene pairs as false positives, each and every predicted isoform of a given fusion gene pair was being called as false positives. For example, in TableS3, Arriba is shown to predict the fusion *LRRFIP2--CAV3* with 7 different breakpoints, and so instead of calling one false positive for that fusion, it's counted 7 times, which seems extreme. Approximately 20% of the false positives for Arriba are attributed to this feature. Hundreds of additional false positives involve gene pairs involving the same gene (self-fused), and so would represent an intra-gene effect and not a fusion pair. Over a hundred involve neighboring genes that presumably involve cis-splicing events. The false positives include the gene pair *KANSL1--ARL17* predicted by Arriba, STAR-Fusion, and FusionCatcher and a well known 'normal' fusion found in about 30% of those with European ancestry due to a natural structural rearrangement (see PMID:28881586). It isn't clear why scFusion is not reporting many these fusions, nor why such fusions would be scored as false positives. All of these issues contribute to scFusion appearing more accurate and the comparators less accurate and are ill justified.

Response: We would like to thank the reviewer for pointing out the problem about the self-fused genes and fusions involving neighboring genes. We now filter all these candidates given by different algorithms. In terms of the *KANSL1-ARL17A*, there are three fusions involving the gene pair *KANSL1* and *ARL17A* with different breakpoints. Two of them have only 2 or 3 supporting cells and do not achieve the significance level. The remaining one is not viewed as a false positive since it has 27 supporting reads in the bulk data (Supplementary Table 9-12). This candidate has a large number of supporting cells and supporting reads. However, since it is located in a multiply-mapped region and scFusion only consider uniquely mapped reads, it is not detected by scFusion.

If we consider fusions with the same gene partners but different breakpoints as one fusion candidate, all algorithms reported fewer fusion candidates in the real datasets (Figure R3a-e). However, scFusion still reported fewer fusions than the bulk methods. Among the 5 real datasets, the numbers of reported fusions in multiple myeloma (MM) dataset decreased the most, since most fusions in MM data involve immunoglobulin genes and fusions involving the same immunoglobulin genes are considered as one candidate. In addition, in our view, fusions of the same gene pairs but with different breakpoints should be viewed as two different fusion candidates. Especially, their breakpoints are often far away from each other (Figure R4). Thus, they are not likely to be caused by minor mapping differences and will be translated to different fusion proteins. One such example is the *IgH-WHSC1* fusion in the multiple myeloma data. In this case, there are two fusions that both involve the *IgH* and *WHSC1* genes, but the two genes are fused at different locations (their breakpoints at the *WHSC1* genes are 3590 bp away from each other). The two fusions are from two different patients and cells with these two fusions belong to different clusters in the tSNE plot (Figure R3f or Figure 7c in the manuscript). In the main figure, we cluster the fusions whose breakpoints are 20 bp away from each other and consider them as one fusion candidate. Fusions of the same gene pairs are considered as separate fusions if their breakpoints are 20 bp away from each other.

Figure R3: The numbers of reported gene fusion in (a) Spike-in, (b) T-cell, (c) MM, (d) LNCaP, and (e) prostate patient data. Fusions involving the same pair of genes are counted only once. (f) The tSNE plot of all MM single cells. The cells with two *IgH-WHSC1* fusions are colored in the plot. The cells from patient RRMM2 and SMM0 are marked by triangle and rectangle, respectively.

Figure R4: The distributions of distances between breakpoints of fusions involving the same gene partners in the experimental spike-in data. Fusions detected by **(a)** Arriba and **(b)** STAR-Fusion.

3. The authors indicate "The reference genome was always chosen as hg19. For the bulk methods, we also applied the same ad hoc filters of scFusion to make different algorithms comparable. Specifically, we applied the pseudogene-, lncRNA-, noapproved-symbol-gene-, intron-, too-many-partner- and too-many-discordant- filters to the fusion candidates of bulk methods." However, many LINC- predictions are apparent in the supplementary tables, indicating that identical filtering rules were not applied as described, and the comparators many false positives were being declared for lncRNAs. I haven't carefully examined whether other filtering criteria were being applied.

Response: Thank you for pointing out this problem. These LINC- predictions are not filtered because of a bug in the filter code. We now fixed the bug and carefully checked other filters and updated all the results in the figures and supplementary tables. The table below shows the numbers of reported fusion before and after the change (self-fused genes, neighboring gene fusions and the filters mentioned in your comments) (Table R1).

Table R1: The numbers of reported fusion before and after the change.

	Spike-in		T-cell		MM		LNCaP		Prostate Patient	
	before	after	before	after	before	after	before	after	before	after
scFusion	53	51	8	8	44	38	5	4	10	10
Arriba	2400	375	457	92	293	41	26	14	192	74
STAR-Fusion	407	310	84	57	144	104	38	36	26	23
FusionCatcher	1931	1124	716	304	167	89	100	73	41	32
EricScript	13262	9044	4306	2884	2160	1492	397	288	162	113

4. The fusion filtering criteria applied by scFusion involving removing lncRNAs and those genes without approved symbols²¹ (such as RP11-475J5.6) do not appear to be justified. If there are true cancer-relevant fusions involving such genes, and there are several well known (PVT1 of PVT1--MYC) or recently discovered, they would be excluded without any scientific basis for doing so.

Response: Figure R5 shows the results without these two filters. We made these two filters optional. Users can choose to turn on or turn off these two filters. Because it is difficult to evaluate the functions of IncRNAs² and genes without approved symbols³, we removed these fusions. Since these filters are applied to all methods, the comparison is fair.

Figure R5: The numbers of reported gene fusions if the no-approved-symbol and IncRNA- filters are not applied in (a) Spike-in, (b) T-cell, (c) MM, (d) LNCaP, and (e) prostate patient data.

5. While it is true that scFusion is the first fusion method that is specifically tailored towards predicting fusions in single cell data, the statistical model component of scFusion is the only method that takes into consideration features of single cell RNA-seq data. This should be made explicitly clear in the manuscript.

Response: We made it clear in the paper that scRNA-seq features are considered by the statistical model and the deep learning model is designed for removing artefacts potentially caused by mis-priming during PCR amplification (see page 9 of the Discussion section).

Minor issues:

6. Figure 1 (a, b) and Figure 1 (c,d,e,f) are disjoint and should be presented separately to avoid conflating aspects of the statistical model and those of the neural network.

Response: We now display them as two separate figures (Figure S1 and Figure S2).

7. While the neural network appears to have good accuracy at discriminating chimeric alignments from others, it also appears to contribute very little to overall fusion prediction accuracy. It makes me wonder if the chimeric alignments are enriched for certain chimeric breakpoints rather than being more generally informative of chimeric alignments. If the chimeric alignments are not first normalized for chimeric breakpoints (such that chimeric breakpoints are more uniformly represented among the large list of chimeric alignments), perhaps this should be performed so that certain breakpoints aren't greatly biasing the predictions, and the neural network may be able to contribute more to discriminating among fusion predictions in scFusion.

Response: Thanks for the suggestion. Although the deep learning model contributes less than the statistical model to the fusion prediction accuracy, it is still an important component of scFusion, as shown in the simulation. In real data analysis, if we turn off the deep learning model, scFusion detects 7 more gene fusions in our spike-in data (Figure R6a). None of the 7 additional fusions are in the 27 spiked-in fusions and none of them have supporting reads in bulk data. Similarly, in the LNCaP dataset, 6 extra fusions are identified by scFusion when the deep-learning model is turned off, but none of them are listed in the CCLE fusion database, and only 40% of the candidate fusions (originally 100%) are in the CCLE fusion database (Figure R6b).

Figure R6: Number of fusion candidates reported by scFusion when turning off the deep-learning model in **(a)** the spike-in data and **(b)** the LNCaP data. In the figure, “Total” means the total number of reported fusions. “Validated” means the number of fusions in the 27 spiked-in fusions or supported by bulk reads. “In CCLE database” means the number of reported fusions listed in the CCLE fusion database of the LNCaP cell line.

One reason that the deep learning model looks less effective is that many of the potential artefacts are also filtered by the *ad hoc* filters (such as the pseudogene-filter and the ≥ 2 cells filter). Figure R7 shows the distribution of artefact scores given by the deep learning model for the fusion candidates before applying these *ad hoc* filters and after applying the *ad hoc* filter. Clearly, before applying the *ad hoc* filters, the scores are more concentrated on the region close to 1, but after applying the *ad hoc* filters, the distribution’s peak around 1 becomes much smaller and we start to see peaks < 0.5 . Hence, in Figure 4b, many potential artefacts have been filtered by the *ad hoc* filters, and it appears that only a minor amount of the potential false positives are filtered by the deep learning model.

Figure R7: The density plots of scores of fusion candidates from (a) LNCaP data and (b) prostate patient data. The red curves are the densities of all candidates from that data, and the blue and green curves are the densities of candidates passed the first filter and the second filter in Figure 4b, respectively.

8. The simulated RNA-seq data used for the initial simulation experiment should be made available in addition to the predictions according to method.

Response: We included the prediction results from one simulation as the supplementary data (Supplementary Table 2-6). The simulated scRNA-seq data was too large (17 terabytes). We made a small part of the simulation data publicly available at <https://drive.google.com/drive/folders/1H5esW31rrq7in4jxBwtGMhJ9v5C81-to?usp=sharing> and provide the source code for generating the simulation data in GitHub <https://github.com/ZijieJin/SimulationscFusion>.

9. The fusion predictions for scFusion in table S2 sometimes have flipped orientation with respect to the spiked in fusions. ie. NUP214--SET and ROS1--GOPC.

Response: We update the output of scFusion to make sure that the first gene is the upstream gene of the gene fusion.

10. For Figure 5C, it isn't clear how the bulk fusions were defined - please include their definition, results in supplementary data, and be sure the rna-seq data are publicly available if not already.

Response: Sorry about the confusion. We did not detect fusions using bulk data. Instead, given a list of fusion candidates detected in the scRNA-seq data, we searched for chimeric reads in bulk data that support the fusion candidates. More specifically, we first mapped all bulk reads to the reference genome using STAR. Then, given any fusion detected in the scRNA-seq data, we searched for chimeric reads in the bulk data whose breakpoints are with 20 bp of the fusion's breakpoints. If a fusion had two or more such supporting chimeric reads, we viewed the fusion as a true discovery. The numbers of supporting reads from bulk data are included in the supplementary tables. The bulk sequencing data is also available at the Genome Sequence Archive (GSA) of BIG Data Center with an accession number HRA001199 (review link <https://ngdc.cncb.ac.cn/gsa-human/s/Kr8lh5>).

11. From my experience, Ericscript has provided little value as a fusion predictor given its high false positive rates. It isn't clear why the authors include Ericscript here as opposed to other tools available.

Response: We included EricScript because EricScript had the best performance in a benchmark article⁴ in 2016.

12. Thanks for making a docker image available. I highly recommend adding your Dockerfile to github - this provides an unambiguous installation protocol for others.

Response: The docker image installation steps and its manual are shown in the “Alternative installation” section in README on GitHub.

To install, users can first run
``docker pull jzj2035198/scfusion``

If the STAR index files are in the XXX/ folder, one can run the following to test scFusion
``docker run -v XXX:/data --rm jzj2035198/scfusion python -u /usr/local/src/scFusion-1.4/scFusion.py -f /usr/local/src/scFusion-1.4/Testdata/ -b 1 -e 10 -s /data/hg19STARIndex/ -t 20 -n 0.9``

13. I did try running your test data through using your docker image and I was not successful.

Response: This problem occurred because the annotation file had several different formats and scFusion only used one of them. To make scFusion more user-friendly, we updated scFusion such that it could handle annotation files in a number of common formats (GENCODE, CCDS, Ensembl, and RefSeq). The README file on GitHub gives detailed guidance about the format of the annotation file required by scFusion.

Reviewer 2:

The resubmission from Jin and colleagues has been revised in response to reviewer comments. Although as stated in my original review, the work is potentially significant, I did not find that the current resubmission, or the responses to my critiques, were adequately addressed.

Response: Thank you for your comments that greatly helped us to improve the manuscript. We misunderstood a few of your comments and thus were not able to adequately address your critiques. We made detailed point-to-point responses below.

(1) Biological significance. As the authors readily admit, the manuscript is purely methodological, the one novel spike-in dataset addressing technical questions. Without a demonstration of how the proposed method detecting fusion transcripts in single-cell RNA sequencing data can drive biological discovery, this study is perhaps better suited for a more technically oriented specialist journal.

Response: We applied scFusion to a prostate data in the revision. scFusion detected 27 cells having the well-known *TMPRSS2-ERG* fusion. Among the 27 cells, 21 are from the patient 1115655. The patient was treated with enzalutamide (an androgen receptor inhibitor). Interestingly, before the enzalutamide treatment, 16.7% (18/108) of cells contain the fusion, much higher than cells after the treatment (3.7% or 3/81), indicating that *TMPRSS2-ERG* fusion might confer efficacy of enzalutamide (Table R2).

Table R2. The contingency table between the *TMPRSS2-ERG* fusion and the enzalutamide treatment. The Fisher's exact test (two-sided) gives a p-value of 0.0047.

	With the fusion	Without the fusion
Before treatment	18	90
After treatment	3	78

In the T cell data, scFusion identified 126 and 20 MAIT cells with the invariant recombinations *TRAJ33-TRAV1-2* and *TRAJ12-TRAV1-2*, respectively. In the literature, it is known that MAIT cells have three invariant recombinations⁵ and that *TRAJ33-TRAV1-2* occurs more frequently than the other two recombinations. However, the relative abundance of the recombinations in MAIT cells is unclear. Here, we provide a direct estimate. The relative abundance of *TRAJ33-TRAV1-2* and *TRAJ12-TRAV1-2* is about 6:1. The remaining *TRAJ20-TRAV1-2* recombination is probably much rarer than *TRAJ33-TRAV1-2* and *TRAJ12-TRAV1-2*, because scFusion did not detect any cells with this recombination. Moreover, we performed differential expression analysis between the cells with the *TRAJ33-TRAV1-2* or *TRAJ12-TRAV1-2* recombination. Interestingly, *ALPK1* and *TIFA* are highly expressed in the cells with *TRAJ12-TRAV1-2* (Figure R8). Recent studies⁶ show that *ALPK1-TIFA* axis is a core innate immune pathway against pathogens such as *Helicobacter pylori*, implying that MAIT cells expressing different TCR α -chains might have different roles in the immune system.

Figure R8: The volcano plot of the differential expression genes of cells between cells with the *TRAJ33-TRAV1-2* or *TRAJ12-TRAV1-2* recombinations. *ALPK1* and *TIFA* are highly expressed in the cells with *TRAJ12-TRAV1-2*.

In the multiple myeloma data, we showed that *WHSC1* expression was dramatically increased immediately downstream of the *IgH-WHSC1* fusion (Figure R9a or Figure 7e in the manuscript), which provided direct evidence that the *IgH-WHSC1* fusion lead to the over-expression of the oncogene *WHSC1*. Differential expression analysis found 115 upregulated genes and 12 down-regulated genes in the cells with the *IgH-WHSC1* fusions (Figure R9b or Supplementary Fig 6d). The upregulated genes include genes known to be co-expressed with *WHSC1* such as *MAL*⁷ and *SCARNA22*⁸. The downregulated genes include known oncogenes in MM such as *CCND1* and *FRZB*, which only expressed in cells without the *IgH-WHSC1* fusion.

Figure R9: (a) The mean read depth of *WHSC1* at different locations for the cells with the two *IgH-WHSC1* fusions and the cells without the fusions. The black triangles indicate the breakpoints of the two fusions. The supporting numbers of splicing junctions are also shown in the plot (the numbers above the arcs). The read depth of a single cell at a location is calculated as the number of reads covering the location per million. The mean read depth is the average depth of all cells in a group. **(b)** The differential expression genes between cells with and without the *IgH-WHSC1* fusion.

(2) *Method descriptions.* Despite claims that the methods have been made more accessible, they are still highly mathematical, without giving an insight why the specific distributions and assumptions are appropriate. For example, the authors write "By assuming that the distribution of the background noises is a zero-inflated negative binomial (ZINB) distribution...". But why is the noise best approximated as a zero-inflated negative binomial?

Response: We are sorry that the method description is still too mathematical. We now provide more explanations about the model setup and model assumptions.

scRNA-seq data are count data and are often modeled by Poisson, negative binomial (NB) or zero-inflated negative binomial (ZINB) distributions⁹⁻¹². The ZINB distribution is the most generalized version of these distributions. Poisson and NB distributions can be viewed as special cases of the ZINB distribution. The NB distribution is more flexible than the Poisson distribution because the NB can allow over-dispersion (i.e., it can allow its variance larger than its mean). The ZINB distribution is more flexible than the NB distribution in that it allows excessive numbers of zeros because of the zero-inflation part of the ZINB distribution. The ZINB model is more appropriate in our context because the data matrix of supporting chimeric reads has an excessive number of zeros (> 95%).

Furthermore, we observed that the candidate fusion's supporting chimeric read numbers strongly depended on the expression of the partner genes and on the local GC content (Figure R10 and Figure 2a, b in the manuscript). To account for this dependency, we employ a regression model. In the regression model, we use two regression functions to describe the dependencies of the two important parameters, the mean and the proportion of zeros, in the ZINB distribution on the gene expression and the local GC content. Since we observed that the gene expression was largely linearly correlated with the supporting read number (Figure R10a and Figure 2a in the manuscript), the regression functions against the gene expression are linear. The GC-content dependency is nonlinear (Figure R10b and Figure 2b in the manuscript) and we use a mathematics technique called splines to represent the nonlinear relationship. By using the splines, we do not need to assume a parametric form for the GC-content dependency (e.g. linear or parabolic or any other form). The spline method is very flexible and can accurately approximate a wide class of smooth functions. Thus, the regression model is also flexible and can describe a wide class of GC-content dependencies.

Figure R10: Features of technical chimeric reads. The number of supporting chimeric reads depends on **(a)** the expression of partner genes and **(b)** the local GC content. The Pearson's Correlations between the number of chimeric reads and the gene expression and the p-values are shown in the figure. The GC content is calculated using sequences near breakpoints (200 bp).

(3) Fusion simulation. In response to my criticism that the fusions that are likely to be biologically present in a sample may not be well represented by randomly putting together reads, the authors address a different problem, i.e. whether the method trained for identifying and rejecting artifactual fusions based on simulated reads also work for identifying and rejecting randomly spliced together "read reads". The original problem still remains unaddressed.

Response: We are sorry that our responses focused on the second half of your comments and missed the first point of your comment. We now redesigned a simulation. Instead of generating fusions by randomly putting together reads, we generated fusions by randomly sampling from the fusions from the Pan-Cancer Analysis of Whole Genomes (PCAWG) study. In this way, we can largely guarantee that the simulated fusions are likely to be biologically present in cancer samples (assuming that most of the PCAWG fusions are true fusions). Other simulation setups are similar as before. Figure R11 or Figure 3 shows the simulation results in this setup. Compared with bulk methods, scFusion had similar levels of recalls but higher levels of precisions and F-scores. For example, in the simulation with 1,000 single cells and 4 million reads per cell, the precision and F-score of scFusion are 0.921 and 0.925, respectively, while the precisions of the bulk methods are only 0.434-0.503 and the F-scores are 0.574-0.667.

Figure R11: The precisions and recalls of scFusion and four bulk methods in six different simulation setups. The figures in the two rows correspond to simulations with 1,000 cells and 500 cells, and the figures in the three columns correspond to simulations with 2 million, 3 million, and 4 million reads in each data. The dots in the figures are the means of precisions and recalls of ten simulations in each setup. The dashed lines are the contour lines with constant F-scores (F-scores are marked in the top-left figure). To illustrate the effect of the deep learning model, this figure also shows the results of scFusion when the deep learning model is turned off (scFusion w/o deep-learning).

(4) *False positive rate assertions.* When addressing the issue of specificity (namely that, just by calling a smaller number of fusions, scFusion has a lower false positive rate compared to other methods), the authors start talking about sensitivity. Again, this does not address the original question.

Response: We are sorry that we did not make the response clear. Since the T-cell data is public data, we cannot use bulk RNA-seq data or use experiment to validate the detected fusions and to estimate the specificity. Instead, we demonstrate that the proportion of true fusions of scFusion is much larger than other methods, thus implying that the false discovery rate of scFusion is lower than other methods. Since these are T cells, the detected V(D)J recombinations are very likely to be true positives. Among the 8 fusions detected by scFusion, 50% are V(D)J recombinations, much higher than other methods (Figure R12a and Figure 6b in the manuscript). Among the V(D)J recombinations detected by scFusion, two are invariant recombinations of MAIT cells. The cells

with these two recombinations highly expressed the MAIT marker gene *SLC4A10* and thus the two recombinations are very likely to be true positives. In comparison, although other methods detected much more fusions, few of the fusions are V(D)J recombinations. Considering that the T-cells are normal cells, other than V(D)J recombinations, they should have very few fusions and thus the majority of the candidate fusions given by the bulk methods are very likely to be false positives.

The T-cell data could not provide an accurate estimation of the specificities of different algorithms. So, we performed a spike-in experiment. In the spike-in experiment, we knew that the spike-in fusions were true positives and we had bulk RNA-seq data to validate the detected fusions in single cells. Thus, sensitivity and specificity could be estimated relatively accurately. Around 80% of the detected fusions were supported by bulk data or were the spike-in fusions, much higher than other methods (Figure R12b). This clearly demonstrated that scFusion had better specificity than other methods.

In addition, we also considered a new dataset from the cancer cell line LNCaP. The CCLE database listed known fusions of this cancer cell line. We could compare the detected fusions with the known fusion list to estimate the false positive rate. scFusion detected 4 fusions and all were in this list. In comparison, < 43% of the fusions reported by bulk methods are in the CCLE database (Figure R12c and Supplementary Figure 7a), again showing that scFusion had a higher specificity.

Figure R12: (a) The proportions of V(D)J recombinations in fusions detected by the five methods in the T cell data. (b) The proportions of fusions having bulk supporting chimeric reads or in the 27 spike-in fusions. (c) The number of detected fusions and fusions in the CCLE fusion database by the five methods in the LNCaP data.

On the positive side, the overall wording of the manuscript is now substantially improved.

Response: Thanks.

Overall, the current revision does not address the critical comments in my original review. Most issues have been sidestepped, or "explained away", rather than addressed, as detailed above.

Response: We highly appreciate your comments that help us to improve our manuscript. We did not mean to sidestep or explain away your critiques. The inadequate responses were mostly due

to our misunderstanding of your comments or our unclear presentations. We provided more detailed responses to address these problems.

Reference

- 1 Haas, B. J. *et al.* Chimeric 16S rRNA sequence formation and detection in Sanger and 454-pyrosequenced PCR amplicons. *Genome Research* **21**, 494-504, (2011).
- 2 Volders, P.-J. *et al.* LNCipedia: a database for annotated human lncRNA transcript sequences and structures. *Nucleic Acids Research* **41**, D246-D251, (2012).
- 3 Braschi, B., Seal, R. L., Tweedie, S., Jones, T. E. M. & Bruford, E. A. The risks of using unapproved gene symbols. *The American Journal of Human Genetics* **108**, 1813-1816, (2021).
- 4 Kumar, S., Vo, A. D., Qin, F. & Li, H. Comparative assessment of methods for the fusion transcripts detection from RNA-Seq data. *Scientific Reports* **6**, 21597, (2016).
- 5 Reantragoon, R. *et al.* Antigen-loaded MR1 tetramers define T cell receptor heterogeneity in mucosal-associated invariant T cells. *J Exp Med* **210**, 2305-2320, (2013).
- 6 Zimmermann, S. *et al.* ALPK1- and TIFA-Dependent Innate Immune Response Triggered by the Helicobacter pylori Type IV Secretion System. *Cell Reports* **20**, 2384-2395, (2017).
- 7 Keats, J. J., Reiman, T., Belch, A. R. & Pilarski, L. M. Ten years and counting: so what do we know about t(4;14)(p16;q32) multiple myeloma. *Leukemia & Lymphoma* **47**, 2289-2300, (2006).
- 8 Mahajan, N., Weber, J. D., Maggi, L. B. & Tomasson, M. H. ACA11, a Small Nucleolar RNA Activated in Multiple Myeloma, Stimulates Proliferation by Inactivating NRF2 and Increasing Redox Signaling. *The FASEB Journal* **30**, 1054.1057-1054.1057, (2016).
- 9 Sun, T., Song, D., Li, W. V. & Li, J. J. scDesign2: a transparent simulator that generates high-fidelity single-cell gene expression count data with gene correlations captured. *Genome Biology* **22**, 1-37, (2021).
- 10 Risso, D., Perraudeau, F., Gribkova, S., Dudoit, S. & Vert, J. P. A general and flexible method for signal extraction from single-cell RNA-seq data. *Nature Communications* **9**, 284, (2018).
- 11 Townes, F. W., Hicks, S. C., Aryee, M. J. & Irizarry, R. A. Feature selection and dimension reduction for single-cell RNA-Seq based on a multinomial model. *Genome Biology* **20**, 1-16, (2019).
- 12 Sarkar, A. & Stephens, M. Separating measurement and expression models clarifies confusion in single-cell RNA sequencing analysis. *Nature Genetics* **53**, 770-777, (2021).

Reviewers' Comments:

Reviewer #1:

Remarks to the Author:

After this second round of revisions, the manuscript continues to be improved. I appreciate the extensive response to the reviewers' critiques in the previous revision, and I appreciate the related updates to the manuscript to address these comments. I do see, however, that not all detailed responses in the response to the last review have been incorporated into updates to the manuscript or the supplementary materials, but rather focused on addressing concerns of the reviewers alone. It would be best to incorporate the R-figures in the response into the supplementary materials so that important concerns are addressed for all readers and not only for the sake of the review process. This is particularly relevant to my concerns relating to the relative importance of the statistical vs. deep learning models being employed by scFusion, and further addressing the issue of how benchmarking considers different breakpoints in the same fusion pair as being distinct true vs. false positive calls.

I appreciate that the authors identified a bug in their latest analysis involving filtering and subsequent scoring of fusion calling accuracy with regard to lncRNAs, and updated the results accordingly. While it doesn't appear to have a major impact on the overall relative standings of different methods, it did appear to substantially impact the reported counts for certain methods.

I don't understand the authors reluctance to add the Dockerfile to the github repo. I personally find it to be the most unambiguous demonstration of the path to installing the software, regardless of whether or not the docker image itself is made available. This is akin to the difference in providing a binary/compiled code vs. providing the open source code, with the latter nowadays being a fairly standard practice for published computational methods.

Reviewer #2:

Remarks to the Author:

In this second revision, the authors finally adequately addressed the critical comments I raised in my earlier reviews. They also added a dataset that demonstrates biological application of the scFusion method. I feel that this revision raised the quality of the manuscript significantly.

Responses to Reviewers' Comments

We list our responses below. The comments are shown in italic and our responses are in blue.

Reviewer 1:

After this second round of revisions, the manuscript continues to be improved. I appreciate the extensive response to the reviewers' critiques in the previous revision, and I appreciate the related updates to the manuscript to address these comments. I do see, however, that not all detailed responses in the response to the last review have been incorporated into updates to the manuscript or the supplementary materials, but rather focused on addressing concerns of the reviewers alone. It would be best to incorporate the R-figures in the response into the supplementary materials so that important concerns are addressed for all readers and not only for the sake of the review process. This is particularly relevant to my concerns relating to the relative importance of the statistical vs. deep learning models being employed by scFusion, and further addressing the issue of how benchmarking considers different breakpoints in the same fusion pair as being distinct true vs. false positive calls.

Response: Thank you for your suggestions. We added Figure R2, R5, R6, and R7 to the supplementary materials (Supplementary Fig. 10, 14, 9, and 11). In addition, we opted for transparent peer review and thus all R-figures should be publicly available.

I appreciate that the authors identified a bug in their latest analysis involving filtering and subsequent scoring of fusion calling accuracy with regard to lncRNAs, and updated the results accordingly. While it doesn't appear to have a major impact on the overall relative standings of different methods, it did appear to substantially impact the reported counts for certain methods.

Response: Thank you again for helping us identify the bugs.

I don't understand the authors reluctance to add the Dockerfile to the github repo. I personally find it to be the most unambiguous demonstration of the path to installing the software, regardless of whether or not the docker image itself is made available. This is akin to the difference in providing a binary/compiled code vs. providing the open source code, with the latter nowadays being a fairly standard practice for published computational methods.

Response: Thanks. We made the Dockerfile and added it into the GitHub repo.

Reviewer 2:

In this second revision, the authors finally adequately addressed the critical comments I raised in my earlier reviews. They also added a dataset that demonstrates biological application of the scFusion method. I feel that this revision raised the quality of the manuscript significantly.

Response: Thank you.